# Genome-wide screens identify SEL1L as an intracellular rheostat controlling collagen turnover

Michael J. Podolsky [1] ✉, Benjamin Kheyfets[1], Monika Pandey[1], Afaq H. Beigh[1], Christopher D. Yang [2], Carlos O. Lizama [2], Ritwik Datta[2], Liangguang L. Lin [3], Zhihong Wang [3], Paul J. Wolters [4], Michael T. McManus [5], Ling Qi [3] & Kamran Atabai [2,4,6] ✉

Accumulating evidence has implicated impaired extracellular matrix (ECM) clearance as a key factor in fibrotic disease. Despite decades of research elucidating the effectors of ECM clearance, relatively little is understood regarding the upstream regulation of this process. Collagen is the most abundant constituent of normal and fibrotic ECM in mammalian tissues. Its catabolism occurs through extracellular proteolysis and cell-mediated uptake of collagen fragments for intracellular degradation. Given the paucity of information regarding the regulation of this latter process, here we execute unbiased genome-wide screens to understand the molecular underpinnings of cell-mediated collagen clearance. Using this approach, we discover a mechanism through which collagen biosynthesis is sensed by cells internally and directly regulates clearance of extracellular collagen. The sensing mechanism appears to be dependent on endoplasmic reticulum-resident protein SEL1L and occurs via a noncanonical function of this protein. This pathway functions as a homeostatic negative feedback loop that limits collagen accumulation in tissues. In human fibrotic lung disease, the induction of this collagen clearance pathway by collagen synthesis is impaired, thereby contributing to the pathological accumulation of collagen in lung tissue. Thus, we describe cell-autonomous, rheostatic collagen clearance as an important pathway of tissue homeostasis.

Fibrosis can be seen as the excess accumulation of extracellular matrix that interferes with tissue and organ function. Collagen is the most abundant constituent of fibrotic extracellular matrix, yet we still lack a complete understanding of the regulatory mechanisms governing collagen homeostasis. Whereas decades of research have elucidated pathways of collagen deposition in normal tissue and in fibrosis, relatively less is understood about collagen clearance. Clearance of extracellular collagen from tissue occurs via two pathways that can act in parallel and sequentially[1]: an extracellular proteolytic process and a process of intracellular uptake and degradation. This latter process is vitally important in controlling normal tissue homeostasis and mitigating fibrosis as demonstrated

[1]Department of Medicine, Weill Cornell Medical College, New York, NY, USA. [2]Cardiovascular Research Institute, University of California, San Francisco, CA, USA. [3]Department of Molecular Physiology and Biological Physics, University of Virginia School of Medicine, Charlottesville, VA, USA. [4]Department of Medicine, University of California, San Francisco, CA, USA. [5]Department of Microbiology and Immunology and UCSF Diabetes Center, University of California, San Francisco, CA, USA. [6]Lung Biology Center, University of California, San Francisco, CA, USA. ✉e-mail: mip9227@med.cornell.edu; kamran.atabai@ucsf.edu

in several recent publications that show the effects of interfering with this process in animal models of fibrosis[2–5]. Although an impaired collagen-degradative environment is a feature of fibrotic diseases such as Idiopathic Pulmonary Fibrosis (IPF), Scleroderma, and cirrhosis[6–13], very little is known about the role of cell-based collagen degradation in human disease.

A key mediator of cell-based collagen clearance is MRC2, a canonical collagen endocytic receptor that binds to and internalizes collagen via a Fibronectin-2 domain, with subsequent lysosomal degradation[14,15]. Earlier studies of this receptor showed that it mitigates lung fibrosis[3] as well as liver and kidney models of fibrosis[16,17]. Our recent work added to this literature by showing that MRC2-mediated collagen clearance is essential for resolution of fibrosis (rather than development of fibrosis) – deletion of MRC2 is sufficient to impair resolution of fibrosis in the single-dose bleomycin mouse model of lung fibrosis[18].

We and others have been studying the process of cellular uptake and degradation of collagen fragments because of the accumulating evidence that this is an important homeostatic mechanism in vivo and controls fibrosis resolution[2,16–19]. Previous attempts at uncovering novel mediators of this process have met with some success, for example uncovering the role of Flotillin proteins or the regulation by MRC2 of matrix proteases[20,21]. However, experimental approaches have either been targeted based on known genes and pathways or have been limited to a subset of potential genes because of limited technical ability to use genome-wide discovery approaches. The advent of high throughput CRISPR-based screens that are done in a pooled format and leverage high throughput sequencing[22] now permits the unbiased evaluation of all potential genes for any role in upstream regulatory pathways that could be contributing to this process. Therefore in this work we started with complementary CRISPRi and CRISPRa screens to address this knowledge gap, followed by mechanistic work to characterize a previously undescribed pathway that controls cell-based collagen clearance.

## Results

We performed genome wide unbiased screens using CRISPR interference (CRISPRi) and CRISPR activation (CRISPRa) to determine genes that regulate cellular uptake of exogenous collagen (Fig. 1). In independent pooled screens, we used CRISPRi to silence every gene in the genome, or CRISPRa to activate every gene in the genome along with use of approximately 10,000 non-targeting guides. We then performed a phenotypic screen for cellular uptake of fluorescent collagen fragments. The screens were carried out independently with entirely separate guide libraries for CRISPRi and CRISPRa. Coverage was at least 350x throughout the screening process for each screen. The screens were performed in U937 cells, a human monocytic leukemia cell line that readily takes up collagen[23] and grows in suspension, which facilitates the use of a large-scale screen platform requiring culturing of hundreds of millions of cells.

We determined multiple genetic regulators of collagen uptake (Fig. 1b, c; Supplementary Data 1). Some of these were previously known to have a role in this process, most importantly *MRC2*, a canonical collagen endocytic receptor. Most of the other candidates were not previously known to play a role in collagen turnover. Importantly, we found a negative correlation when comparing phenotype scores for individual genes that were hits in both the CRISPRi and CRISPRa screens (Fig. 1d). This suggests that at least some of these genes we identified are both necessary and sufficient for regulating cellular collagen clearance and that quantitative control of collagen clearance by these genes might be tunable based on their expression levels. To validate our findings, we re-screened the high and low uptake bins from the original CRISPR screens individually and compared hits in these re-screens to the corresponding original CRISPRi or CRISPRa screen (Supplementary Fig. 1a–d). We found a marked correlation between the original and repeat screens (Supplementary Fig. 1c, d). To validate the performance of our pooled guide library, we used arrayed delivery of CRISPRi reagents for 3 sgRNAs for multiple genes identified as hits in either positive or negative directions in the original CRISPRi screens, followed by collagen uptake assays. Phenotypic effects of

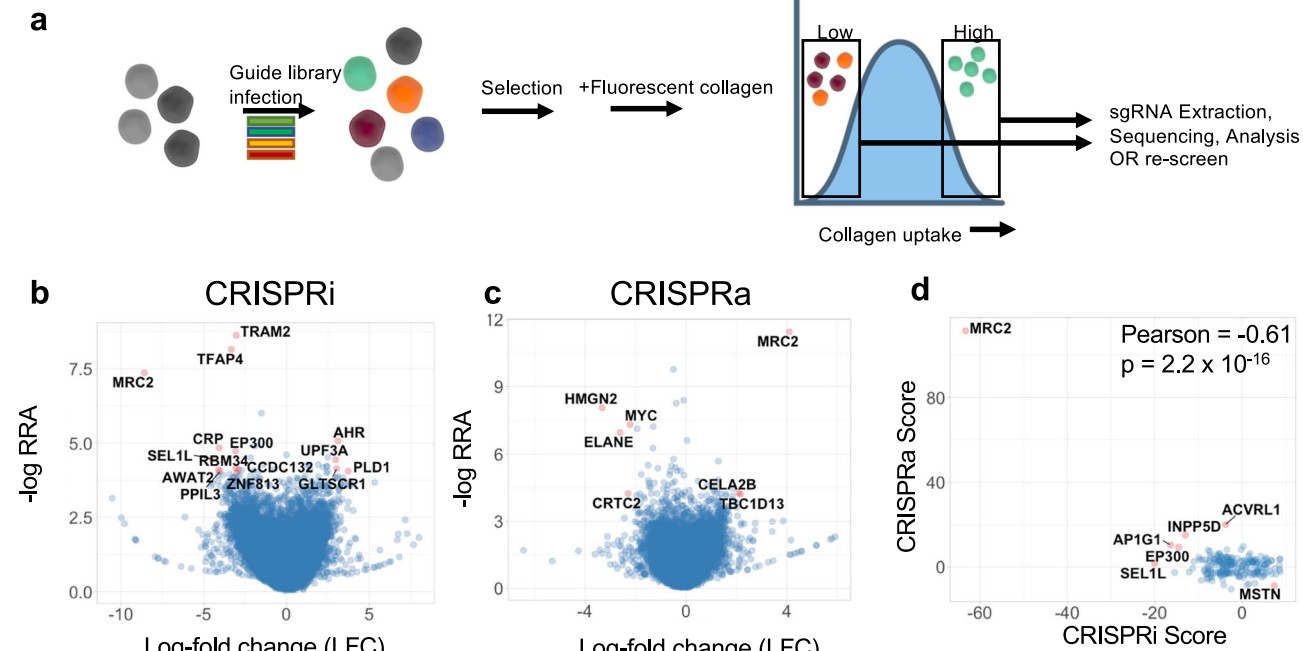

**Fig. 1 | CRISPR screens identify mediators of cellular collagen uptake.**
**a** Schematic of experimental approach for flow cytometry-based screening for uptake of collagen fragments. **b** Volcano plot of gene-level statistics of CRISPRi (inhibition) screen. Most significant gene hits are labeled and colored in red. RRA Robust Rank Aggregation score, LFC Log-fold change. **c** Volcano plot of gene-level statistics of CRISPRa (activation) screen. **d** Comparison of overall phenotype score between CRISPRi and CRISPRa screens; $p = 2.2 \times 10^{-16}$. Statistics: **d** Pearson's correlation, two-sided. Source data are provided in Supplementary Data 1.

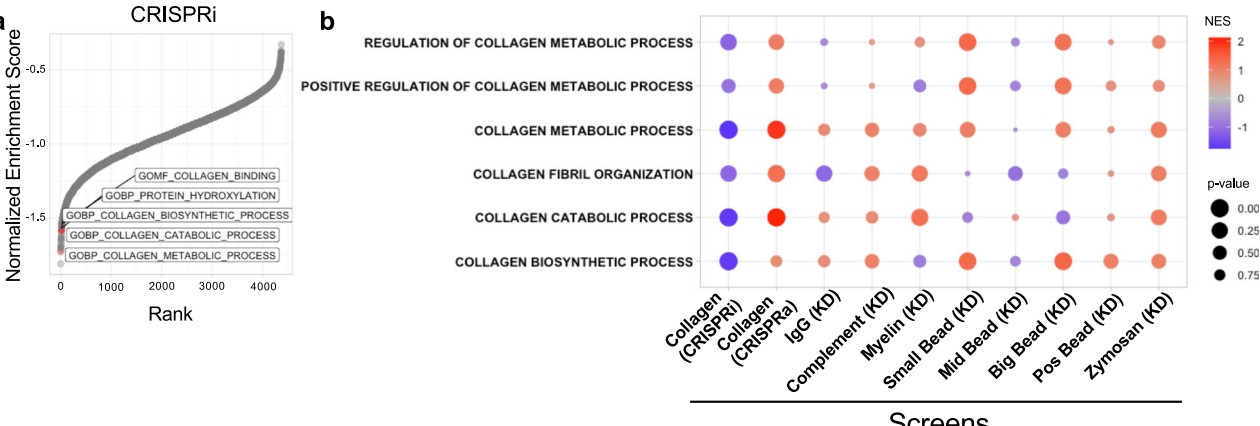

**Fig. 2 | Collagen-biosynthetic pathways are regulators of collagen uptake uniquely compared to screens for uptake of other cargoes. a** Ranked plot of Normalized Enrichment Scores derived from Gene-Set Enrichment Analysis (GSEA) of CRISPRi phenotype scores, using 10,561 Gene Ontology gene sets. Collagen-related gene sets are labeled and displayed in red. **b** Comparison of GSEA statistics for labeled collagen-related gene sets across screens for uptake of various cargoes[26]; screen method is indicated in parentheses. KD Knock-down. Statistics: GSEA algorithm (based on the Kolmogorov–Smirnov test), as described in the Methods section; unadjusted *p*-values are shown. Source data are provided as a Source Data file.

arrayed delivery of these CRISPRi sgRNAs were 77% concordant with original screen data (Supplementary Fig. 1e).

We next performed bioinformatic analyses of hits from our CRISPR screens. We noticed that multiple gene ontology (GO) groups related to collagen metabolism were top scoring GO terms in our analysis using Gene-Set Enrichment Analysis (GSEA; Fig. 2a). This was true in all four screens, i.e. the original CRISRPi and CRISPRa screens as well as the re-screens (Supplementary Fig. 2). Intriguingly, this did not only include collagen catabolism GO terms, but also collagen biosynthesis or collagen production-related GO terms (Fig. 2a). This informatic analysis implied that collagen biosynthetic activity may positively regulate cellular uptake of extracellular collagen fragments. MRC2-mediated collagen uptake occurs via receptor-mediated endocytosis. Collagen uptake can also occur via phagocytosis[24] or macropinocytosis[25], and impairments in any of these processes due to gene silencing could potentially be detected in our screens. We therefore compared the enrichment of these collagen-related GO terms in our collagen uptake screens with data from previously published CRISPR screens of phagocytosis of other cargo[26] (Fig. 2b). Only our screens for collagen uptake exhibited consistent enrichment of collagen biosynthetic GO terms (i.e. positive regulation of collagen uptake by collagen biosynthesis), compared with uptake of other cargo (Fig. 2b). These data suggest that there is a biological basis for positive regulation of collagen uptake by collagen biosynthesis genes, and this is specific to uptake of collagen.

To validate whether collagen biosynthesis positively regulates collagen uptake, we decided to evaluate individual collagen biosynthesis genes. A heatmap of gene-level phenotype scores from our original screens among top-enriched genes from the Collagen Biosynthesis GO group is shown in Fig. 3a. We also note that *TRAM2*, a gene previously described to be critically important collagen biosynthesis[27] was a top hit in the CRISPRi screen as well (Fig. 1). We used an orthogonal technique (shRNA) to silence *TRAM2* in U937 cells which resulted in decreased collagen uptake (Fig. 3b; Supplementary Fig. 3a) and decreased MRC2 expression at the protein and mRNA levels (Fig. 3c, d). To verify whether *TRAM2*-silencing indeed affects collagen synthesis, we used MRC5 human fetal lung fibroblasts which express high levels of collagen. Silencing of *TRAM2* in fibroblasts led to decreased endogenous levels of collagen (Fig. 3e–g) and decreased collagen uptake in fibroblasts as well (Fig. 3h). We next silenced individual collagen genes that were seen in the original screen including type I collagens and type V collagens and determined that silencing of these genes led to decreased collagen uptake and a decrease in *MRC2*

message in U937 cells (Fig. 3i–k). Silencing of type I collagen synthesis in MRC5 fibroblasts also caused decreased collagen uptake (Fig. 3l) as well as decreased MRC2 protein levels (Fig. 3m, n). By contrast, stimulating collagen expression by pharmacologic means with TGFβ1 increased collagen uptake in U937 cells (Fig. 4a). Although it is controversial whether leukocytes or myeloid cells produce significant quantities of collagen[28–30], they do generate collagen mRNA and recently were found to contribute to collagen content in vivo under certain conditions (including types I and IV collagens in zebrafish and mouse)[31]; a Western blot verified that U937 cells do indeed make collagen, albeit in very low quantities (Fig. 4a; note 60 µg of protein needed to be loaded to visualize collagen from U937 cells, compared with 20 µg loaded in most other Western blots in this manuscript; additionally, a more sensitive total type I collagen antibody was used in this subpanel rather than a procollagen antibody that is less sensitive but is specific for newly synthesized collagen, which we used in the other subpanels of this figure). To corroborate these findings, we additionally stimulated collagen synthesis in MRC5 fibroblasts and measured collagen uptake and *MRC2* expression. Stimulation with TGFβ1 led to increased collagen uptake and increased *MRC2* expression (Fig. 4b, c). Using orthogonal pharmacologic stimulators of collagen synthesis, ascorbic acid and LPA1 respectively, led to increasing *MRC2* message as well (Fig. 4d, e). These data together provide additional evidence for a pathway by which collagen biosynthesis regulates *MRC2* expression and collagen uptake.

Because the screens were based on flow cytometric sorting of individual cells and each cell had a single individual genetic perturbation, phenotypes determined by our screens are expected to be fundamentally cell-autonomous. Therefore we presumed that collagen synthesis in cells must be sensed by some internal cell-autonomous mechanism. In examining hits in the screen to determine whether there could be a cell-autonomous sensor of collagen synthesis, we were intrigued to see that *SEL1L* was a top hit in our screens (Fig. 1). The gene product of *SEL1L* is a single-pass, endoplasmic reticulum-resident transmembrane protein with its luminal face containing a Fibronectin-2 (FN2) domain (Fig. 5a). FN2 domains are generally known to bind collagen[32–34]. However, the function of the FN2 domain in SEL1L has not previously been characterized.

To validate the effect of *SEL1L* on collagen uptake, we used lentiviral shRNA-mediated gene silencing in MRC5 lung fibroblasts. Silencing of *SEL1L* led to decreased collagen uptake as well as MRC2 protein (Fig. 5b–d; Supplementary Fig. 3b). As a complementary approach, we also used an inducible knockout mouse embryonic

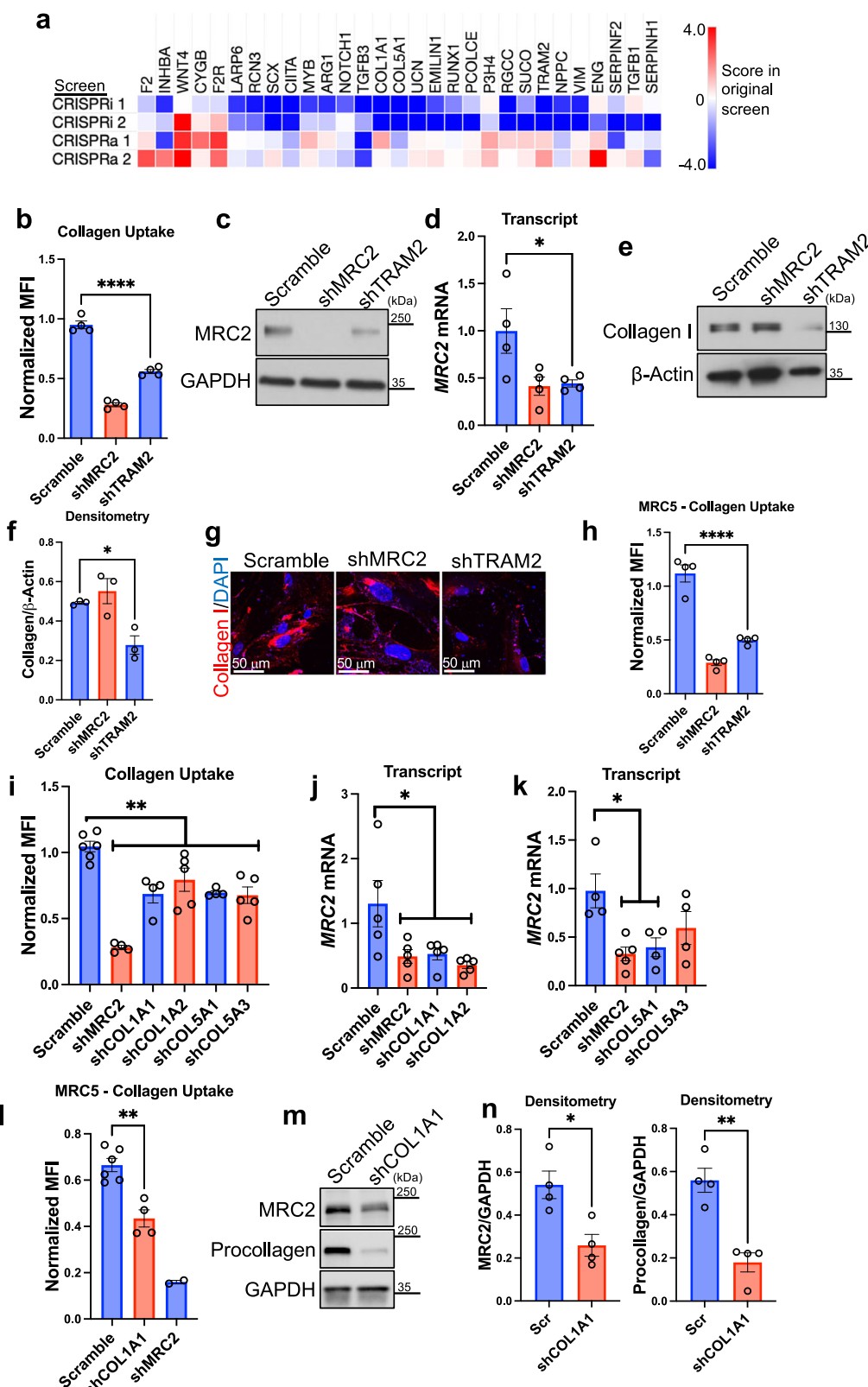

fibroblast (MEF) cell line characterized previously[35]. We determined that when *Sel1L* is deleted, there is reduced collagen uptake (Fig. 5e), reduced MRC2 protein expression (Fig. 5f–g; Supplementary Fig. 3c, d), reduced *Mrc2* mRNA (Fig. 5h; Supplementary Fig. 3e), and reduced MRC2 cell surface and total protein expression (Fig. 5i, j). By contrast overexpressing SEL1L in fibroblasts leads to increased *MRC2* message and protein (Fig. 5k, l). Importantly although SEL1L has chiefly been

described as a member of the endoplasmic reticulum-associated protein degradation (ERAD) quality control pathway[35–37], other members of the ERAD pathway were not hits in our CRISPR screens of collagen uptake (Fig. 1; Supplementary Data 1). This included HRD1, which acts with SEL1L and is essential for ERAD[38–41] but was not a hit in our screens. To test if SEL1L-mediated regulation of collagen uptake is ERAD-dependent, we used cells null for SEL1L or HRD1 and determined that

**Fig. 3 | Collagen biosynthesis positively regulates cell-based collagen clearance and *MRC2* expression. a** Heatmap of overall phenotype scores from original screens as indicated for top genes from 'Collagen Biosynthetic Process' GO gene-set (displayed genes had most extreme 'Score' values in one or more screen result, with most extreme 'Score' absolute value at least >1 and concordant with GO term; 29/43 genes from this GO term that were included in the screen fit these criteria and are displayed). **b** Flow-cytometry based collagen-uptake assay in U937 cells after shRNA treatment as indicated vs. Scramble control. MFI Mean Fluorescence Intensity. $N = 4$ per group; $p < 0.0001$. **c** Western blot of U937 cells after shRNA treatment as indicated vs. Scramble control, representative of $N = 4$ independent experiments. **d** Q-RT-PCR in U937 cells after shRNA treatment as indicated vs. Scramble control. $N = 4$ per group; $p = 0.0483$. **e, f** Western blot and densitometry of MRC5 fibroblasts after shRNA treatment as indicated vs. Scramble control, representative of $N = 3$ independent experiments; $p = 0.0477$. **g** Representative confocal immunofluorescence images of MRC5 fibroblasts after shRNA treatment as indicated vs. Scramble control. **h** Flow-cytometry based collagen-uptake assay in MRC5 cells after shRNA treatment as indicated vs. Scramble control. $N = 4$ per group; $p < 0.0001$. **i** Flow-cytometry based collagen-uptake assay in U937 cells after

shRNA treatment as indicated vs. Scramble control. $N = 6$ (Scramble), 4 (shMRC2, shCOL1A1, shCOL5A1), 5 (shCOL1A2, shCOL5A3) independent biological replicates; $p$-values are for post-hoc testing of Scramble vs. the following: shMRC2 ($p < 0.0001$), shCOL1A1 ($p = 0.0005$), shCOL1A2 ($p = 0.0088$), shCOL5A1 ($p = 0.0006$), shCOL5A3 ($p = 0.0001$). **(j–k)** Q-RT-PCR in U937 cells after shRNA treatment as indicated vs. Scramble control. $N = 5$ (left, all groups), 4 (right, Scramble, shCOL5A1, shCOL5A3), 5 (right, shMRC2) independent biological replicates; **j** $p = 0.0136$ (Scramble vs. shMRC2), 0.0189 (Scramble vs. shCOL1A1), 0.0041 (Scramble vs. shCOL1A2); **k** $p = 0.0080$ (Scramble vs. shMRC2), 0.0223 (Scramble vs. shCOL5A1). **l** Flow-cytometry based collagen-uptake assay in MRC5 fibroblasts after shRNA treatment as indicated vs. Scramble control. $N = 6$ and 4 in Scramble and shCOL1A1 groups respectively, $N = 2$ in shMRC2 group; $p = 0.0011$. **m, n** Western blot and densitometry of MRC5 fibroblasts after shRNA treatment as indicated vs. Scramble control, representative of $N = 4$ independent experiments; $p = 0.0137$ (left), $p = 0.0017$ (right). Data are shown as the mean ± SEM. Statistics: **b–k** one-way ANOVA with post-hoc Dunnett's testing; **(l–n)** unpaired Student's *t* test (two-sided). *$p < 0.05$, **$p < 0.01$, ****$p < 0.0001$. Source data are provided as a Source Data file.

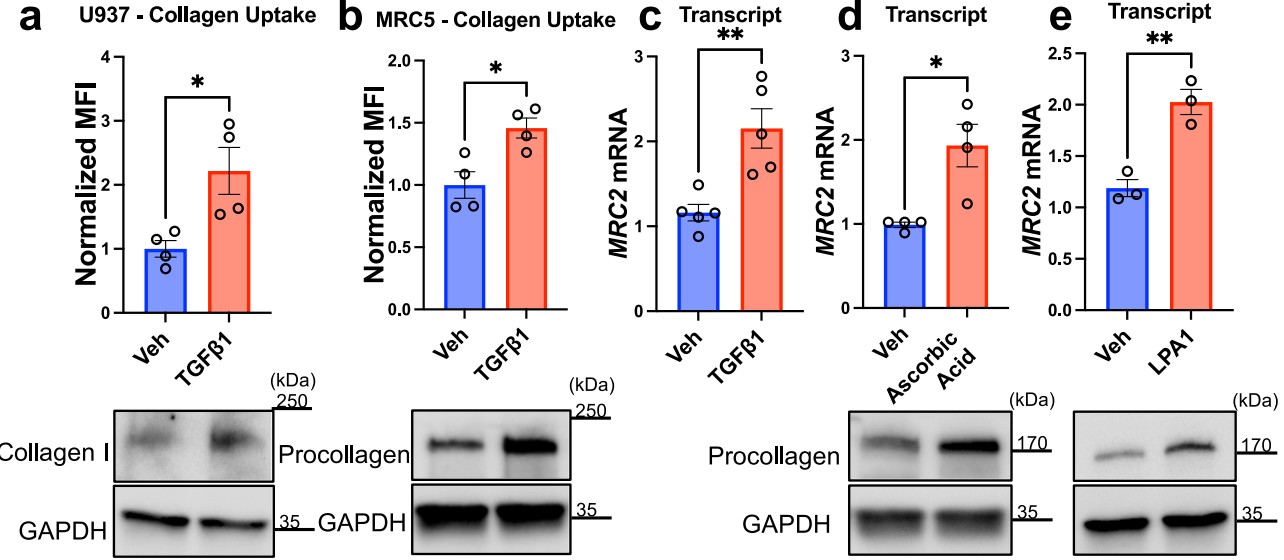

**Fig. 4 | Pharmacological stimulation of collagen synthesis is positively associated with collagen uptake and *MRC2* expression.** Flow-cytometry based collagen-uptake assay in U937 (**a**) or MRC5 (**b**) cells after drug treatment as indicated vs. vehicle control. $N = 4$ per group; $p = 0.0205$ (**a**), $p = 0.0138$ (**b**). Western blot of corresponding cells and conditions shown below, each representative of $N = 3$-4 independent experiments. Q-RT-PCR in MRC5 cells after drug treatment as

indicated vs. vehicle control. $N = 5$ (**c**), 4 (**d**), 3 (**e**) independent experiments; $p = 0.0043$ (**c**), $p = 0.0102$ (**d**), $p = 0.0048$ (**e**). Western blot of corresponding cells and conditions shown below, each representative of $N = 3$ independent experiments. Data are shown as the mean ± SEM. Statistics: **a–e** unpaired Student's *t* test (two-sided). *$p < 0.05$, **$p < 0.01$. Source data are provided as a Source Data file.

only SEL1L positively regulates collagen uptake, MRC2 expression (Fig. 6a–c; Supplementary Fig. 3f), and *MRC2* message production (Fig. 6d). For these experiments and several following cell biology experiments, HEK293T cells were chosen for the following reasons: (1) efficiency of transfection and co-transfection; (2) conserved positive regulation of MRC2 by SEL1L in this cell type as well; and (3) decreased proliferation of MEFs after two passages following induction of complete *Sel1L* knockout, as described previously[35]. However, where possible we have analyzed more than one cell type as described below.

We then tested whether MRC2 can be forcibly overexpressed in wildtype (WT), SEL1L-knockout and HRD1-knockout cells. We were able to overexpress in all three conditions to equal levels MRC2 that is functional and can take up exogenous fluorescent collagen fragments (Fig. 6e–g; Supplementary Fig. 3g). Since HRD1 is essential for ERAD as above, these data suggest that the ERAD pathway is not necessary for MRC2 protein expression or maturation and argues against ERAD itself being instrumental in regulating MRC2 expression. Furthermore, our data indicate that *MRC2* is being regulated by SEL1L at the message level based on the data in Figs. 5 and 6, rather than protein level as may

be expected if the process were ERAD-dependent. Finally, though deficiency of SEL1L or HRD1 have both been shown to predispose to ER stress[42,43], the mechanism by which deficiency of either of these molecules predisposes to ER stress has been shown to be ERAD-dependent. Since the data in Figs. 5 and 6 show that HRD1 deficiency does not affect MRC2 expression and collagen uptake, this argues against a strong contribution of ER stress from SEL1L deficiency affecting this pathway via ERAD impairment because HRD1 deficiency would be expected to have the same effect on ER stress.

To determine whether SEL1L could be a collagen sensor, we wanted to investigate whether it binds collagen and whether its FN2 domain is important for regulating *MRC2* levels. Computational modeling predicted docking of collagen peptide in the FN2 domain of SEL1L (Supplementary Fig. 4a). In fibroblasts, we determined that SEL1L and collagen co-localize, and this colocalization increases when collagen synthesis is induced by TGFβ1 (Fig. 7a; Supplementary Fig. 4b). Furthermore, there is a biochemical association between SEL1L and type I collagen, as determined by co-immunoprecipitation followed by Western blotting (Fig. 7b). We next made expression

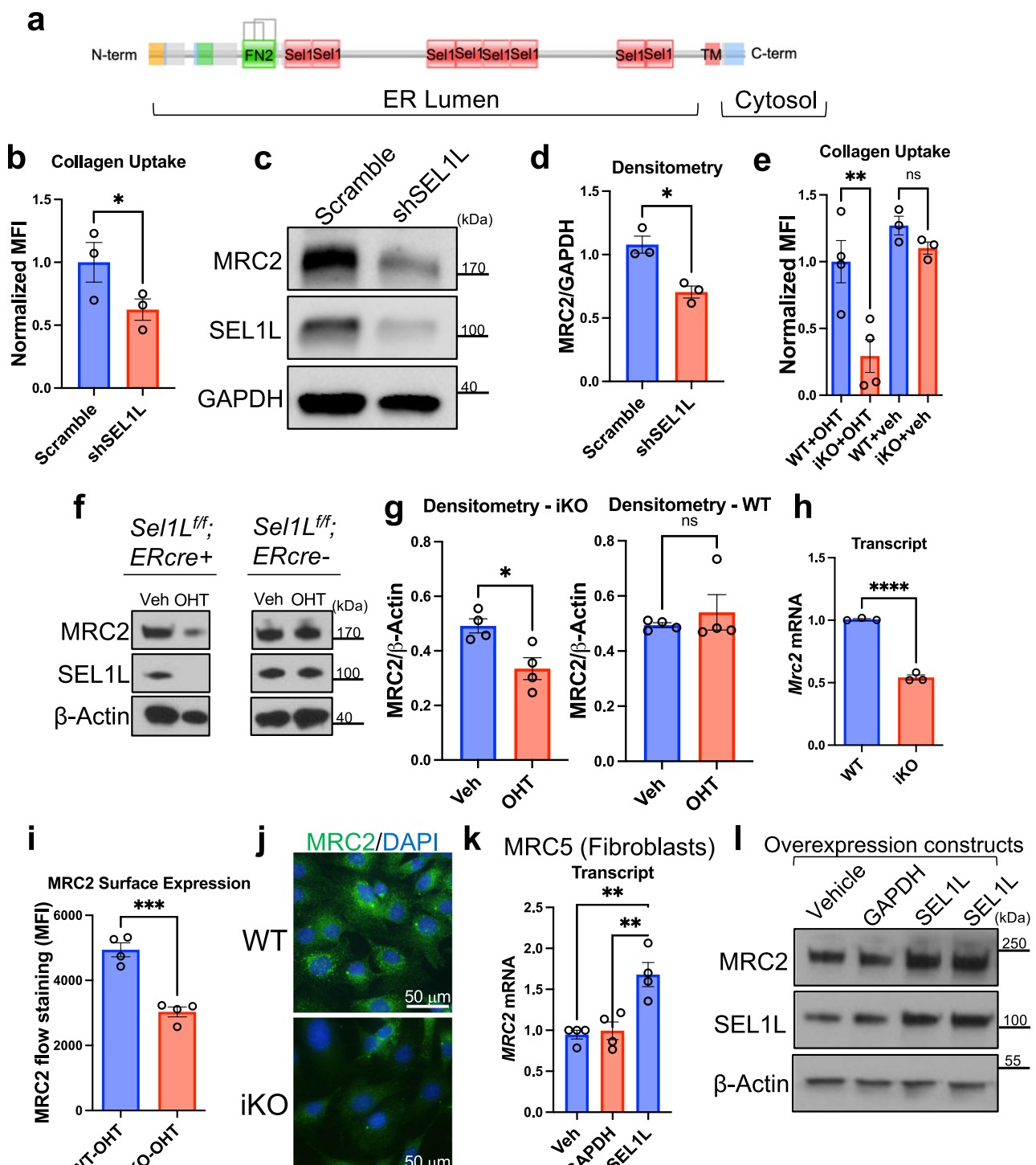

**Fig. 5 | *SEL1L* positively regulates collagen uptake and *MRC2* expression.**
**a** Schematic of domain structure of SEL1L protein. FN2 = Fibronectin-2. **b** Flow-cytometry based collagen-uptake assay in MRC5 lung fibroblasts after treatment with lentiviral-mediated shRNA against SEL1L or Scramble control. $N = 3$ per group, $p = 0.0359$. **c, d** Western blot and densitometry of MRC5 lung fibroblasts after shRNA treatment as indicated vs. Scramble control, representative of $N = 3$ independent experiments; $p = 0.0102$. **e** Flow-cytometry based collagen-uptake assay in MEF cells (mouse embryonic fibroblasts) after treatment with 4-hydroxytamoxifen (OHT) as indicated vs. vehicle control. WT = *Sel1L^{f/f};ERcre-*, iKO = *Sel1L^{f/f};ERcre+*. $N = 4$ (OHT condition), 3 (veh condition) per group; $p = 0.0018$ (left); $p = 0.7830$ (right). **f, g** Western blot and densitometry of MEFs after OHT treatment as indicated vs. control, representative of $N = 4$ independent experiments; $p = 0.0168$ (left), $p = 0.6857$ (right). **h** Q-RT-PCR in MEFs after OHT treatment for genes as

indicated. $N = 3$ per group; $p < 0.0001$. **i** Flow-cytometry based measurement of MRC2 surface expression in MEFs after OHT treatment. $N = 4$ per group; $p = 0.0004$. **j** Representative wide-field immunofluorescence images of MEFs after OHT treatment. **k** Q-RT-PCR of MRC5 cells transfected with overexpression vectors containing genes as indicated vs. vehicle control (lipofectamine only). $N = 4$ per group; $p = 0.0017$ (vs. Veh), $p = 0.0029$ (vs. GAPDH). **l** Western blot of MRC5 cells transfected with overexpression vectors containing genes as indicated vs. vehicle control (lipofectamine only). $N = 2$. Data are shown as the mean ± SEM. Statistics: **b** paired Student's *t* test (two-sided); (**d, g** - left panel, **h, i**) unpaired Student's *t* test (two-sided); (**e, k**) one-way ANOVA with post-hoc Tukey testing; (**g** - right panel) unpaired Mann–Whitney *U* test (two-sided). *$p < 0.05$, **$p < 0.01$, ***$p < 0.001$, ****$p < 0.0001$, ns not significant. Source data are provided as a Source Data file.

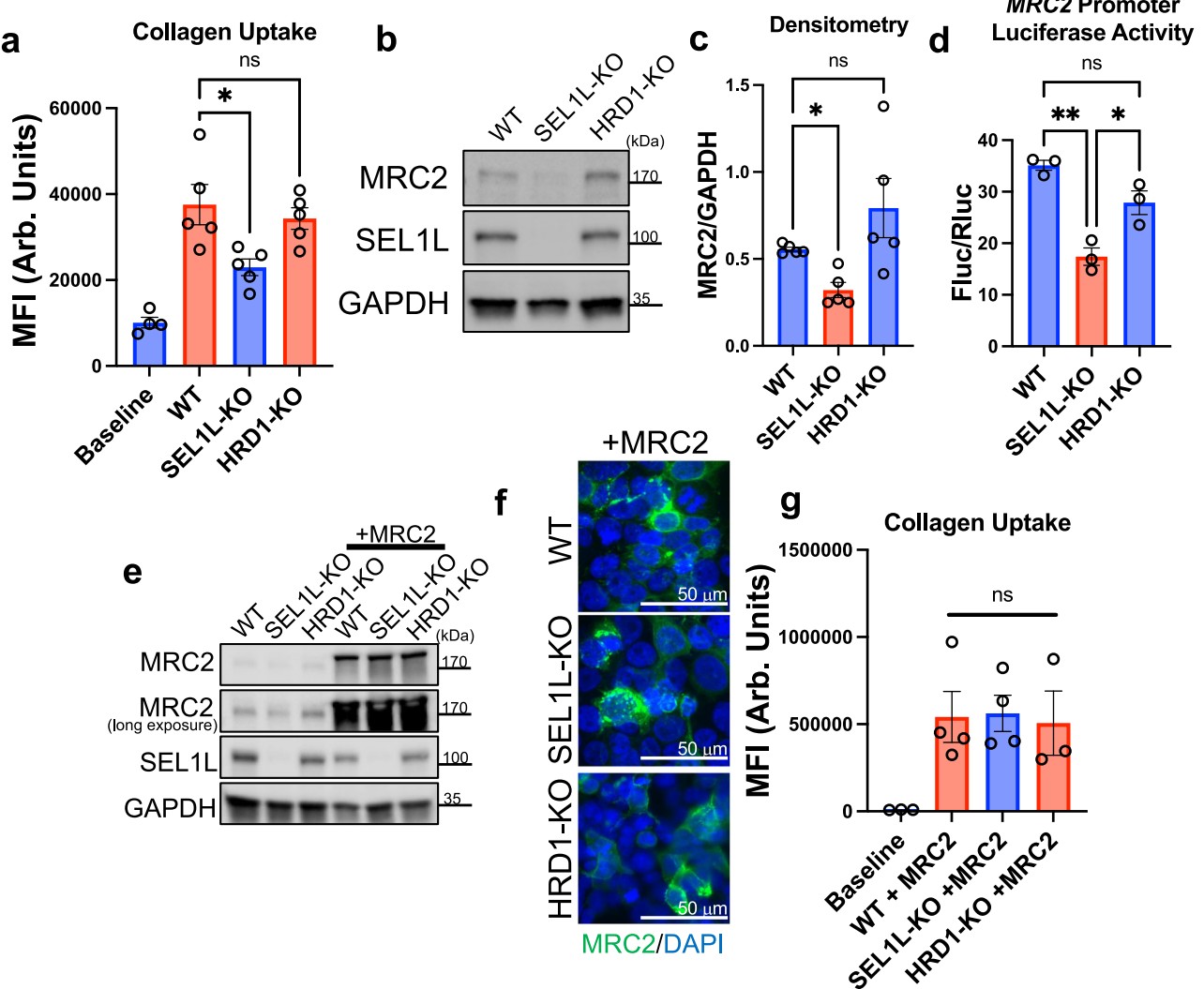

**Fig. 6 | SEL1L, but not HRD1, positively regulates MRC2. a** Flow-cytometry based collagen-uptake assay in HEK293T of different genotypes (compared with baseline control without fluorescent collagen). $N = 4$ (Baseline), 5 (WT, SEL1L-KO, HRD1-KO) per group; $p = 0.0209$ (vs. SEL1L-KO), $p = 0.7682$ (vs. HRD1-KO). **b, c** Western blot and densitometry of HEK293T cells of different genotypes, representative of $N = 5$ independent experiments; $p = 0.0120$ (WT vs. SEL1L-KO), $p = 0.5143$ (WT vs. HRD1-KO). **d** Firefly luciferase assay of *MRC2* promoter activity, normalized to constitutive Renilla luciferase in HEK293T cells of different genotypes. $N = 3$ per group; $p = 0.011$ (WT vs. SEL1L-KO), $p = 0.0162$ (SEL1L-KO vs. HRD1-KO), $p = 0.078$ (WT vs. HRD1-KO). **e** Western blot of HEK293T cells of different genotypes transfected with MRC2 overexpression vector vs. vehicle control, representative of $N = 6$ independent

experiments. **f** Representative wide-field immunofluorescence images of HEK293T cells of different genotypes transfected with MRC2 overexpression vector vs. vehicle control. **g** Flow-cytometry based collagen-uptake assay in HEK293T of different genotypes transfected with MRC2 overexpression vector (compared with baseline control without fluorescent collagen; note different scale compared with panel a). $N = 3$ (Baseline), 4 (WT, SEL1L-KO, HRD1-KO) per group; $p = 0.9993$ (WT vs. SEL1L-KO), $p = 0.9898$ (SEL1L vs. HRD1-KO). Data are shown as the mean ± SEM. Statistics: **a, g** one-way ANOVA with post-hoc Tukey testing; **c** repeated-measures ANOVA with post-hoc Bonferroni testing; **d** one-way ANOVA with post-hoc Bonferroni testing. *$p < 0.05$, **$p < 0.01$, ns not significant. Source data are provided as a Source Data file.

constructs of WT SEL1L vs. SEL1L with an in-frame deletion of its FN2 domain (AA118-166; referred to as ΔFN2), each with C-terminal MYC and FLAG tags, to examine the function of the FN2 domain. Computational prediction suggested that this deletion should have minimal overall effect on protein folding (Supplementary Fig. 5a), since predicted full-length or ΔFN2 structures had very high predicted alignment with Root Mean Squared Distance <1 Å. Transfection of either construct led to localization of the MYC-tagged protein in the ER based on co-localization with Calnexin (Supplementary Fig. 5b). We also attempted purification of the WT and ΔFN2 proteins which migrated to the expected sizes via SDS-PAGE (Supplementary Fig. 5c, d). Other proteins were also pulled down via this method, but the WT and ΔFN2 were enriched compared to whole-cell lysate. The identities of indicated bands in the Supplementary Fig. 5c were confirmed to be SEL1L via proteomic analysis of excised gel bands (Supplementary Data 3; in

proteomics, WT and ΔFN2 could not be differentiated due to lack of unique peptide identifications). Two bands seen on Coomassie staining may represent glycosylated and non-glycosylated SEL1L. We analyzed the eluates from the purification with circular dichroism (Supplementary Fig. 5e). Their spectra were similar, suggesting similar secondary structures of the mix of eluted proteins from both WT and ΔFN2 condition. Taken together, these data suggest the ΔFN2 mutant is fully translated, can be localized in the correct subcellular compartment, and likely folds properly.

Next, we compared co-immunoprecipitation of the WT or ΔFN2 constructs with collagen and show that the SEL1LΔFN2 mutant exhibited diminished binding to procollagen compared to WT SEL1L in lysates from co-transfected cells (Fig. 7c, d). To evaluate this interaction further and to exclude the possibility that collagen synthesis in these cells influences the diminished binding, we incubated our

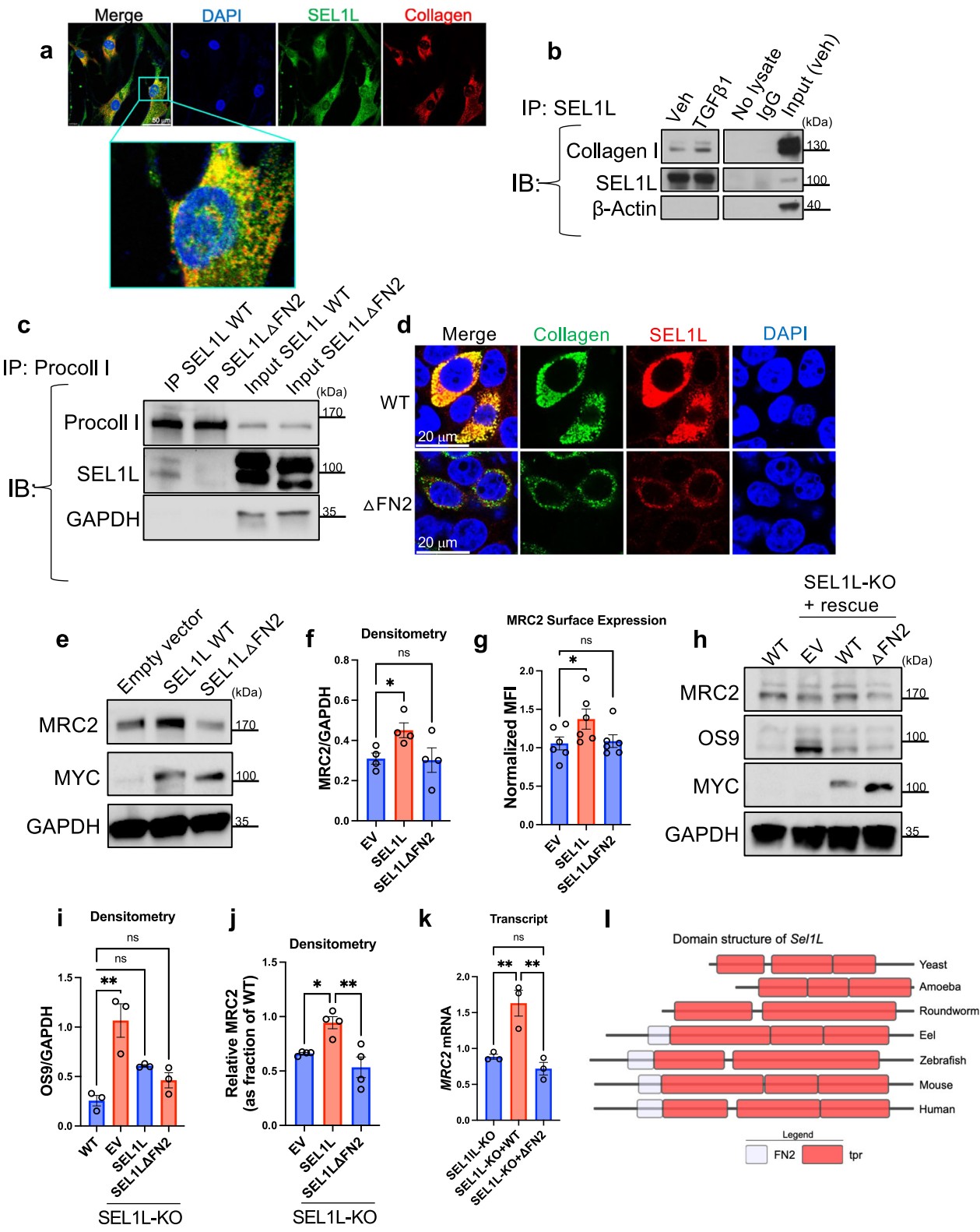

purified protein constructs with rat-tail collagen and demonstrate reduced binding efficiency of the ΔFN2 construct by co-IP, down to the background levels of non-specific binding of collagen to beads (Supplementary Fig. 5f). Since we could not achieve purity of the SEL1L proteins via this anti-MYC purification approach, we employed an additional technique and attempted purification of the WT and mutant SEL1L proteins via anti-FLAG pull-down (Supplementary Fig. 5g; second band running lower than WT SEL1L is of unclear significance; may

represent non-glycosylated SEL1L, or another protein that co-purifies with SEL1L), resulting in overall improved purity although still relatively low yield. Again, WT SEL1L bound less to exogenously added rat-tail collagen in cell-free conditions than did ΔFN2, after using the anti-FLAG for purification followed by co-IP via anti-MYC (Supplementary Fig. 5h). These data collectively indicate that the FN2 domain is essential for the association of SEL1L with collagen either through direct binding of SEL1L and collagen or through a complex of proteins

**Fig. 7 | SEL1L binds collagen as it is being synthesized via its FN2 domain and the FN2 domain is necessary for positive regulation of *MRC2* expression by SEL1L. a** Representative confocal immunofluorescence image of MRC5 fibroblasts, along with magnified inset to highlight co-localization, representative of $N = 4$ independent experiments. **b** Western blot of MRC5 fibroblasts after immunoprecipitation with anti-SEL1L antibody (vs. controls as indicated), representative of $N = 2$ independent experiments. **c** Western blot after immunoprecipitation with anti-procollagen type I antibody (with input controls as indicated) of HEK293T cells transfected with collagen and WT or mutant SEL1L constructs as indicated, representative of $N = 3$ independent experiments. **d** Representative confocal immunofluorescence images of HEK293T cells transfected with collagen and WT or mutant SEL1L constructs as indicated, representative of $N = 3$ independent experiments. **e, f** Western blot and densitometry of MEFs after treatment with WT or mutant SEL1L constructs as indicated, representative of $N = 4$ independent experiments; $p = 0.0305$ (EV vs. SEL1L), $p > 0.9999$ (EV vs. ΔFN2). **g** Flow-cytometry based measurement of MRC2 surface expression on MEFs after treatment with WT or mutant

SEL1L constructs as indicated vs. Empty Vector (EV). $N = 6$ per group; $p = 0.0244$ (EV vs. SEL1L), $p = 0.8913$ (EV vs. ΔFN2). **h–j** Western blot and densitometry of HEK293T cells either WT or KO for SEL1L as indicated after treatment with EV (empty vector), WT or mutant SEL1L constructs as indicated, representative of $N = 3$ (**i**), 4 (**j**) independent experiments; $p = 0.0011$ (**i**, WT vs. EV), $p = 0.0985$ (**i**, WT vs. SEL1L), $p = 0.5120$ (**i**, WT vs. ΔFN2), $p = 0.0241$ (**j**, EV vs. SEL1L), $p = 0.0028$ (**j**, SEL1L vs. ΔFN2). **k** Q-RT-PCR of SEL1L-KO HEK293T cells after transfection with WT or mutant SEL1L as indicated vs. empty vehicle control. $N = 3$ per group; $p = 0.0097$ (KO vs. WT), $p = 0.0036$ (WT vs. ΔFN2), $p = 0.6021$ (KO vs. ΔFN2). **l** Comparison of protein domain structures of Sel1L orthologs across phylogenetically diverse organisms; FN2 Fibronectin-2 domain, tpr Tetratricopeptide repeat domain. Data are shown as the mean ± SEM. Statistics: **f** repeated-measures one-way ANOVA with post-hoc Bonferroni testing; **g, k** one-way ANOVA with post-hoc Tukey testing; **i, j** one-way ANOVA with post-hoc Bonferroni testing. \*$p < 0.05$, \*\*$p < 0.01$, ns not significant. Source data are provided as a Source Data file.

pulled down by SEL1L. We cannot conclude from our data that there is a direct molecular interaction between the FN2 domain and collagen since the purity of SEL1L proteins in the MYC or FLAG-tag based purification was not complete (see Supplementary Fig. 5c and Supplementary Fig. 5g).

Importantly in both fibroblasts and HEK293T cells, WT SEL1L drives MRC2 expression but SEL1LΔFN2 does not, even though in these experiments these expression constructs are expressed at similar levels (Fig. 7e–k). Furthermore, we evaluated the levels of OS9, a well-described ERAD substrate[35], under conditions of SEL1L knockout or rescue with WT or ΔFN2 constructs (Fig. 7h–j). OS9 accumulated in the SEL1L knockout cells as expected, which was rescued with the WT construct as well as the ΔFN2 mutant construct. In sum these data suggest that SEL1L is an internal sensor of collagen biosynthesis and that SEL1L is necessary for the effect of collagen biosynthesis on *MRC2* message levels and hence the downstream phenotype of collagen uptake. Consistent with our data in Fig. 6 suggesting this phenomenon is ERAD-independent, the FN2 domain appears to be essential for positive *MRC2* regulation but not for ERAD based on the effects on OS9 levels. This effect on OS9 also validates the stability and function of the ΔFN2 construct since it rescues ERAD functionality. Consistent with our data suggesting that the FN2 domain is dispensable for ERAD, a comparison of protein domain structures of SEL1L across diverse species demonstrates that although SEL1L and its ERAD function are evolutionarily ancient and conserved down to yeast[44], the FN2 domain is only present in vertebrates and higher organisms, consistent with the hypothesis that the collagen-sensing function arose later and is likely to be independent of its original ERAD function (Fig. 7l). This occurred evolutionarily after the appearance of the earliest collagens in sponges at the dawn of metazoa[45–47].

To further test whether the FN2-dependent regulation of MRC2 by SEL1L could be operating through ERAD or not, we conducted additional experiments (Fig. 8). Using cycloheximide, we showed that OS9 degradation was indeed impaired in SEL1LKO cells as compared with WT (Fig. 8a), as expected. To see if OS9 degradation was affected under conditions of FN2 deletion, we used HEK293T cells in which the FN2 domain was deleted in-frame in the endogenous locus, to avoid any issues with gene overexpression confounding the results. This construct indeed led to diminished MRC2 protein (Fig. 8b; note in Fig. 8b the difference in mobility between the FN2 deletion and WT, expected to be ~50AA different, appears small likely because this is a gradient gel that resolves molecular weight differences relatively less at higher molecular weights), also as expected based on our other data. OS9 degradation rates, however, were not altered in these cells (Fig. 8c). These data suggest, along with our data above, that the FN2 domain may be dispensable for ERAD function, but is key for positively regulating MRC2. Finally, we compared both MRC2 levels and OS9 levels among WT, SEL1LKO, and the in-frame ΔFN2 mutant cells after

treatment with cycloheximide, proteasome inhibitor MG-132, or vehicle controls (Fig. 8d, e). OS9 degradation is impaired in the SEL1LKO cells, but not WT and ΔFN2 cells, and proteasome inhibition appears to mitigate the degradation of OS9 somewhat in WT and ΔFN2 cells. However, there is no effect on OS9 levels after treatment with these drugs compared to the baseline condition in SEL1LKO cells, consistent with the impaired ERAD function in SEL1LKO cells, but not WT and ΔFN2 cells. Interestingly, MRC2 levels follow a relatively similar pattern. We interpret these data to mean that MRC2 could possibly be an ERAD substrate (though this is not proven with these data alone), but that the similar effect of downregulation of MRC2 levels seen in both SEL1LKO cells and ΔFN2 cells is therefore likely a result of ERAD-independent processes – presumably, at the transcript level according to our data in Fig. 7. To validate this conclusion further, we tested whether the deletion of the FN2 domain has any effect on ERAD by evaluating additional known endogenous ERAD substrates (Fig. 8f), including CD147[48], SHH[49], and IRE1α[50]. In all instances, SEL1LKO had the expected effect of leading to impaired degradation of these substrates after cycloheximide treatment whereas deletion of the FN2 domain had no effect, consistent with similar results recently reported by our co-authors[51].

To determine whether this pathway is relevant in human physiology and disease, we re-analyzed single cell data from multiple published experiments including normal and fibrotic lung. We found that in cells that express both *MRC2* and type I collagen, there is a positive correlation between expression levels of these two mRNA species (Fig. 9a–d). However, this positive correlation is diminished IPF lung cells (Fig. 9d). In fact, data from multiple independent diverse datasets examining gene expression in different compartments (whole lung, bronchoalveolar lavage, peripheral blood mononuclear cells) from control versus IPF patients demonstrate that in all cases there is a positive correlation between collagen and *MRC2* expression, but the relationship is impaired in IPF (Fig. 9e–g). These findings corroborate our in vitro data showing collagen biosynthesis is positively associated with *MRC2* and collagen uptake by cells. This diminished recruitment of *MRC2* could be a potential mechanism by which fibrotic collagenous extracellular matrix fails to be cleared in diseases of persistent and progressive fibrosis like IPF. By contrast in a spontaneously resolving model of fibrosis (the bleomycin lung fibrosis model in C57Bl\6 mice) there is a positive correlation between collagen and *Mrc2* expression in mouse lung and the relationship is not diminished under conditions of fibrosis (Fig. 9h). Given that we have previously shown that MRC2 is necessary for resolution of fibrosis[18], these data raise the possibility that impairment of biosynthesis-induced collagen clearance via MRC2 could be an important pathway driving non-resolving fibrosis and IPF.

Finally, we examined SEL1L levels in lungs of pulmonary fibrosis patients versus control lungs (see Supplementary Data 4 for more details). We found that in some, but not all, fibrotic human lung

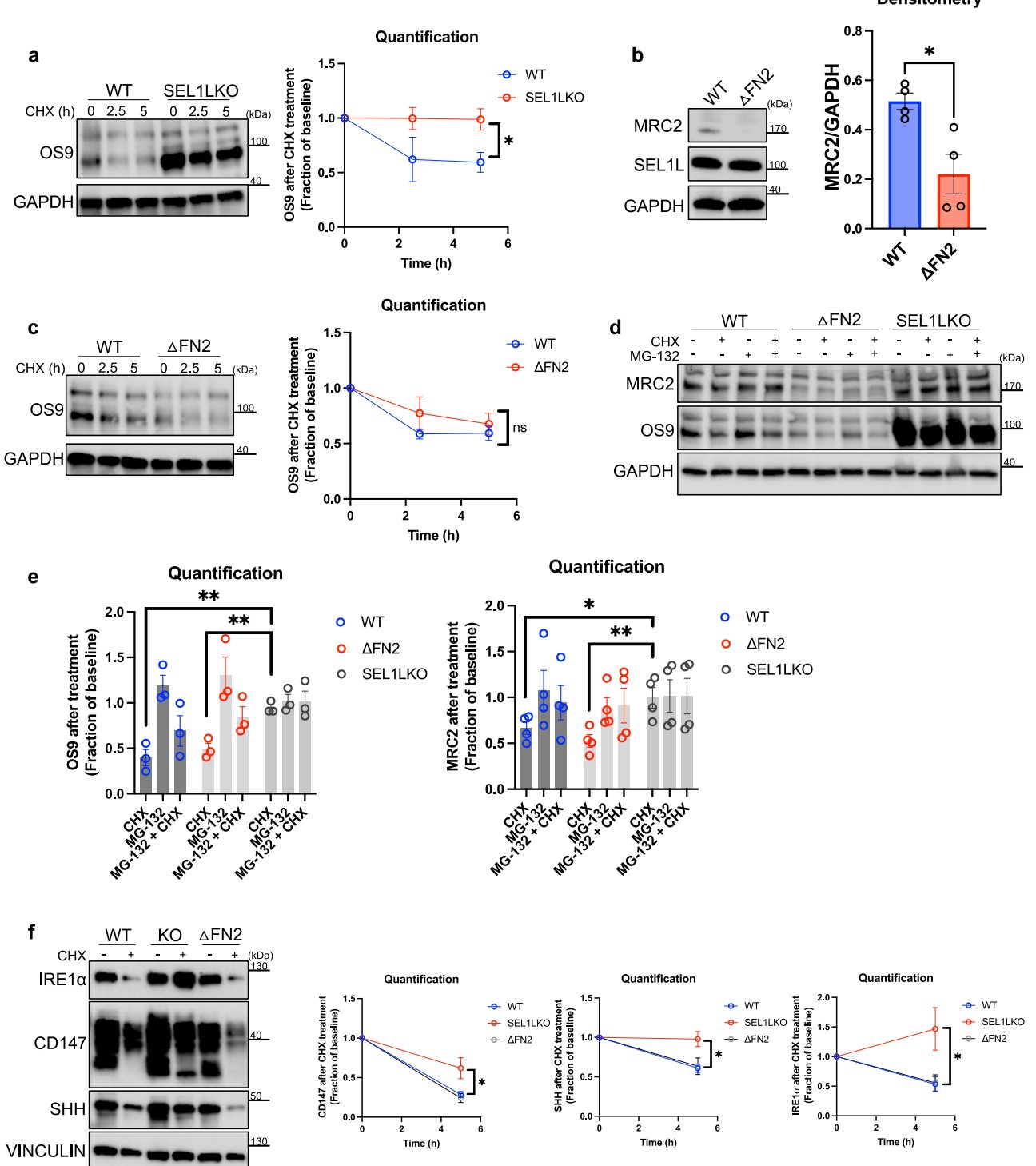

specimens there is diminished full-length SEL1L (Fig. 10a, b). Immunofluorescence in fibrosis vs. control lungs corroborated this finding (Fig. 10c). Future work will investigate how this process is taking place. Additional mechanisms by which collagen biosynthesis-induced collagen clearance is impaired in IPF is also an area of study in our laboratory.

## Discussion

In this work we have shown that collagen synthesis directly upregulates collagen clearance by MRC2. Our screens identified uniquely important biological pathways controlling collagen clearance that do not regulate phagocytic uptake of other cargo. This led us to identify SEL1L as a critical regulator of collagen synthesis-induced collagen turnover. We have presented several lines of evidence that suggest SEL1L is an internal sensor of collagen biosynthesis and that its collagen sensing function, via a FN2 domain, mediates the homeostatic effect of collagen biosynthesis on upregulation of MRC2. Whether SEL1L directly senses collagen through its FN2 domain or rather is part of a protein complex that senses collagen is difficult to discern based on our data though the known binding of FN2 domains to collagen[32–34] argues in favor of direct binding. SEL1L is known to be a key member of the ERAD pathway, but our data indicate that the collagen-sensing function of SEL1L is independent of its ERAD function, defining a noncanonical function for this protein. Consistent with this concept is

**Fig. 8 | The FN2 domain of SEL1L is likely dispensable for ERAD. a** Western blot and quantification (normalized to baseline for each condition) of WT vs. SEL1LKO cells for indicated proteins after treatment with cycloheximide for time specified, representative of $N = 3$ independent experiments; $p = 0.0428$ (at 5 h). **b** Western blot and densitometry of WT vs. ΔFN2 HEK293T cells created via CRISPR-mediated deletion of FN2 domain for indicated proteins, representative of $N = 4$ independent experiments; $p = 0.0145$. **c** Western blot and quantification (normalized to baseline for each condition) of WT vs. ΔFN2 cells as described in panel **b** for indicated proteins after treatment with cycloheximide for time specified, representative of $N = 3$ independent experiments; $p = 0.400$ (at 5 h). **d, e** Western blot and quantification (expressed as fraction of vehicle only treated baseline for each condition) of WT vs. ΔFN2 vs. SEL1LKO cells for indicated proteins after treatment with cycloheximide and/or proteasome inhibitor MG-132, representative of $N = 3$ (left), 4

(right) independent experiments; $p = 0.0022$ (left, KO vs. WT), $p = 0.0058$ (left, KO vs. ΔFN2), $p = 0.0467$ (right, KO vs. WT), $p = 0.0075$ (KO vs. ΔFN2). **f** Western blot and quantification (expressed as fraction of vehicle only treated baseline for each condition) of WT vs. SEL1LKO vs. ΔFN2 cells for indicated proteins after treatment with cycloheximide, representative of $N = 4$ independent experiments; quantification of CD147 refers to the lower band (core-glycosylated CD147); all $p$-values at 5 h: $p = 0.0468$ (CD147, KO vs. WT), $p = 0.0258$ (CD147, KO vs. ΔFN2), $p = 0.0309$ (SHH, KO vs. WT), $p = 0.0431$ (SHH, KO vs. ΔFN2), $p = 0.0393$ (IRE1α, KO vs. WT), $p = 0.0440$ (IRE1α, KO vs. ΔFN2). Data are shown as the mean ± SEM. Statistics: **a, b** unpaired Student's $t$ test (two-sided); **c** unpaired Mann–Whitney $U$ test (two-sided). *$p < 0.05$. **e, f** one-way ANOVA with post-hoc Bonferroni testing. *$p < 0.05$, **$p < 0.01$, ns not significant. Source data are provided as a Source Data file.

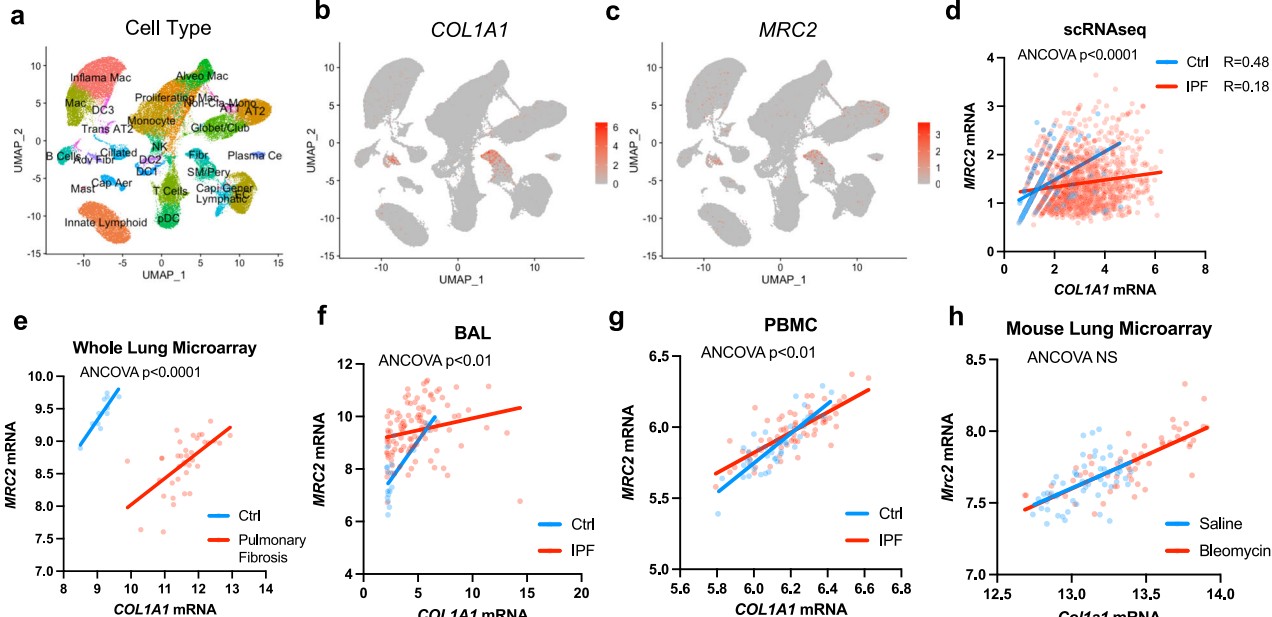

**Fig. 9 | Collagen biosynthesis and *MRC2* expression are positively correlated in lung tissue, but this relationship is impaired in human fibrotic lung disease. a** UMAP plot with cell type annotation of scRNAseq data from multiple available datasets of human lung control and IPF tissue (GSE136831, GSE135893, GSE121611, GSE128033, GSE132771). $N = 184,672$ single cells. **b, c** Gene expression of individual genes as indicated in UMAP as generated in panel **a**. **d** Comparison of gene expression in all cells from data as in panel **a** that had non-zero expression of *COL1A1* or *MRC2*; $p < 0.0001$. **e** Comparison of gene expression in whole human lung microarray data from GSE110147 analyzed via GEO2R in pulmonary fibrosis lungs vs. control specimens. $N = 37$ vs. 11 per group; $p < 0.0001$. **f** Comparison of microarray gene expression in human lung bronchoalveolar lavage specimens from

GSE70867 analyzed via GEO2R from Idiopathic Pulmonary Fibrosis (IPF) lungs vs. control specimens. $N = 112$ vs. 20; $p = 0.0016$. **g** Comparison of microarray gene expression in peripheral blood mononuclear cell (PBMC) specimens from GSE37858 analyzed via GEO2R from Idiopathic Pulmonary Fibrosis (IPF) lungs vs. control specimens. $N = 70$ vs. 35; $p = 0.0083$. **h** Comparison of gene expression in whole mouse lung microarray data from GSE40151 analyzed via GEO2R from Bleomycin-treated mice vs. Saline control mice. $N = 56$ vs. 55; $p = 0.9189$. Statistics: ANCOVA of two groups (no multiple comparisons correction), $p$-values indicated in figures. Pearson correlations are displayed in panel **d**. Source data are provided as a Source Data file.

our observation that although SEL1L and the ERAD pathways are both evolutionarily ancient with conservation to yeast[44], the collagen-sensing FN2 domain of SEL1L arose more recently with vertebrates after the evolutionary origin of collagen proteins, which appeared concurrently with metazoa[45–47].

From a teleological perspective, priming the same cells that contribute to scar formation to upregulate machinery integral for repair and clearance of that scar would be an efficient means of reaching a goal of tissue homeostasis and regeneration of normal tissue. In IPF, a disease of persistent non-resolving fibrosis, the relationship between collagen synthesis and clearance is uncoupled, in contrast to the preserved relationship of these two processes in the spontaneously resolving model of bleomycin-induced lung fibrosis in C57Bl\6 mice[18,52,53]. These findings suggest this pathway could be ripe for therapeutic intervention in IPF. Until now, it has not been apparent that this rheostatic collagen turnover pathway was inadequately

engaged because previous studies have shown a modest increase in *MRC2* expression in IPF lung compared with normal lung[54–58]. However, this belies an inappropriate recruitment of the homeostatic turnover mechanism necessary to resolve fibrosis which would be expected commensurate with high levels of collagen synthesis in IPF.

In short, an imbalance of collagen production and degradation is necessary for the hallmark accumulation of collagen in pulmonary fibrosis; our work here suggests a defective cell-mediated pathway of collagen clearance, based on expression of MRC2 which correlates closely to functional uptake of collagen[23], is a key part of this imbalance. It is well documented that MRC2 mitigates fibrosis in in vivo models[3,16,17]. Direct cellular clearance of collagen by MRC2 is likely the key function responsible for this phenotype. Collagen fragments are also thought to be pro-inflammatory[59,60], so it is possible that uncleared collagen fragments in the setting of inadequate MRC2 activity could add to fibrogenesis. However, MRC2 has multiple other

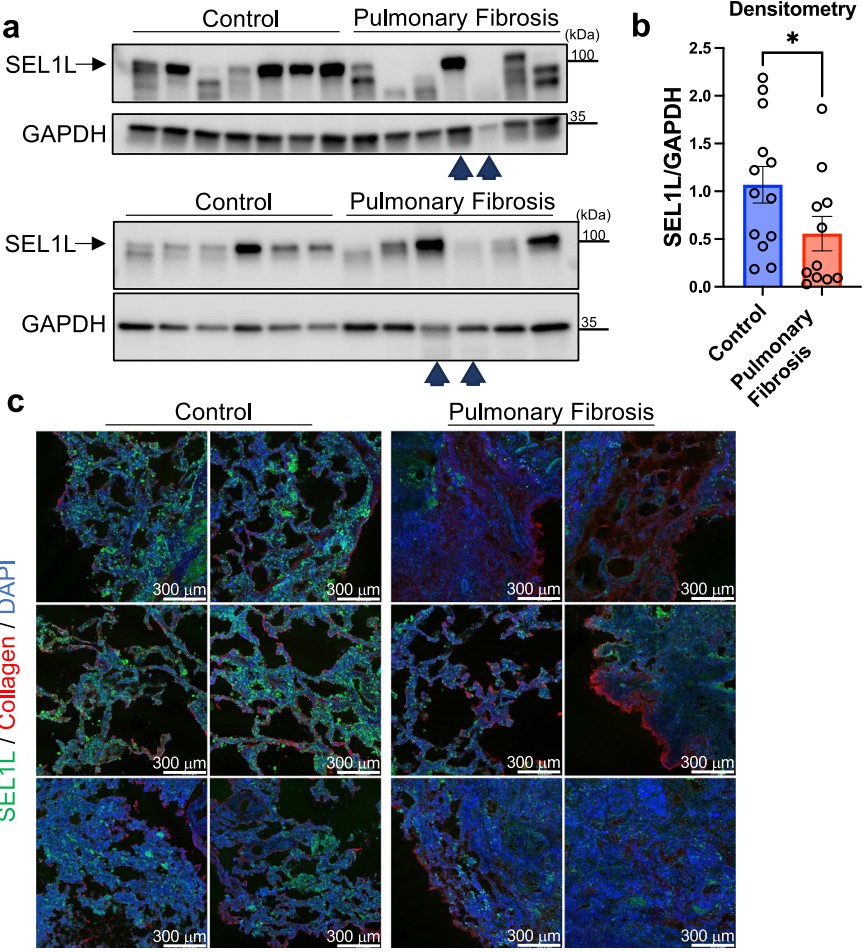

**Fig. 10 | SEL1L protein levels are reduced in the lungs of patients with Pulmonary Fibrosis. a**, **b** Western blots of whole human lung lysates from Pulmonary Fibrosis lungs vs. control (donor lungs deemed not suitable for transplant), with corresponding densitometry. $N = 13$ in Control and 11 in Pulmonary Fibrosis group. Arrowheads correspond to two lung lysates that were used in both Western blots for normalization across blots, otherwise all other lanes are independent samples; $p = 0.0352$. **c** Representative low-power wide-field immunofluorescence images of human lungs (Pulmonary Fibrosis vs. control). Each row comes from an individual donor or Pulmonary Fibrosis lung, with two different fields per lung (i.e. $N = 3$ individual lungs per group). Data are shown as the mean ± SEM. Statistics: **b** unpaired Mann–Whitney $U$ test (two-sided). *$p < 0.05$. Source data are provided as a Source Data file.

functions besides collagen uptake such as clearance of collectins[61] and roles in apoptosis regulation and other cellular processes[62,63]. The inadequate upregulation of MRC2 we have described in pulmonary fibrosis could have multiple downstream consequences which we intend to explore further in future work. Despite this, our gain-of-function data showing that overexpression of SEL1L can lead to increased MRC2 production suggests this pathway could be therapeutically targeted (i.e. to drive increased clearance of collagen to ameliorate fibrosis). The decrement in SEL1L seen in a subset of fibrotic lungs in both Western blotting and immunofluorescence lends further credence to this idea, since MRC2 is pro-resolving in fibrosis[18].

Limitations of our study include not elucidating the specific cell type(s) in which SEL1L is apparently diminished in pulmonary fibrosis lungs; this is an area of future study in our laboratory. SEL1L is expressed in multiple cell types in the lung and may have other roles (most notably its function in ERAD) that are or are not relevant in fibrosis biology. Since collagen expression can be detected in other cell types besides fibroblasts, it is possible the pathway we have described here is operational in other cell types as well. We also do not yet understand how SEL1L is downregulated in fibrotic human lungs. Our data suggest this is one potential mechanism by which there is incomplete recruitment of the homeostatic collagen clearance mechanisms in IPF. How SEL1L itself is downregulated in fibrotic tissue is also the subject of ongoing study in our laboratory. In addition, the

mechanism by which SEL1L-mediated collagen sensing transcriptionally regulates *MRC2* is not completely determined by this work. The luciferase activity data we have produced suggest that *MRC2* transcript production may be a downstream effect of SEL1L expression levels. Our data suggest that MRC2 may also be an ERAD target, though we have not proven this here; however, the mechanism by which SEL1L positively regulates *MRC2* mRNA we have shown here is apparently not ERAD-dependent. Lastly, though our data suggest the FN2 domain could act to bind collagen, we cannot exclude that there is an additional player or players mediating the interaction between SEL1L and collagen, which would require additional molecular studies.

Despite these limitations, we demonstrate here that unbiased genome-wide approaches based on phenotypic screening can yield new insights into cell and tissue-level biology. The relationship between matrix production and degradation is turning out to be ever more complex; we believe understanding how this core relationship is dysregulated in pulmonary fibrosis at the cellular level has the potential to help unlock insights about this relentlessly progressive disease.

## Methods
### Study approval
All experiments using human tissue were approved by the Weill Cornell Medicine or UCSF Committee on Human Research. Informed consent was obtained for study participants or waived for human

tissue from deceased donors in accordance with the UCSF Committee on Human Research.

## CRISPR screens

Single cell clones of U937 cells expressing dCas9-KRAB-BFP (CRISPRi) or dCas9-VP64-BFP (CRISPRa) were made as previously described[64]. Subsequently, independent guide libraries for CRISPRi and CRISPRa were used as previously described[65]. Briefly, lentivirus particles containing these guides were transduced into U937 clones above at low MOI (0.3), followed by subsequent selection with puromycin (1 μg/ml). Puromycin-selected cells were amplified and then underwent the fluorescent collagen uptake assay (as below) in bulk (i.e. pooled format), with sorting into high and low bins (top or bottom 10% of collagen uptake based on fluorescence intensity), simultaneously on 4x FACSAria II devices, then re-cultured and harvested for analysis or for repeat screening. Isolation of cell DNA, followed by amplification of the guide-region of individual cells and high-throughput amplicon sequencing on an Illumina platform were all done as previously described[65,66].

## Bioinformatics, including analysis of screen data

Guide counts were extracted using Python v3 scripts previously published[22]. Analysis of screen data was done via MAGeCK 0.5.9[67] using the lfc-mean method. Bioinformatic analysis was performed using GSEA 4.2.3[68,69] on phenotype scores (defined as Log-fold change multiplied by the -$\log_{10}$(p-value) as determined by the robust rank-aggregation method in MAGeCK. For comparison across knockdown screens for different cargo, data from Haney MS et al. were used[26] and then re-analyzed via GSEA along with our own collagen screens. Visualization of screen data was done with ggplot2 package in R (version 4.1.1). Visualization of domain architecture of SEL1L protein across species was performed using the OMA Orthology Browser[70]. For re-analysis of single cell RNAseq data from multiple experiments (GSE136831, GSE135893, GSE121611, GSE128033, GSE132771) were integrated and re-normalized via the Seurat R package version 4; cells with non-zero expression of *COL1A1* and *MRC2* were used for analysis in Fig. 9d. Microarray data was re-analyzed as indicated using GEO2R (R 4.2.2, Biobase 2.58.0, GEOquery 2.66.0, limma 3.54.0).

## Quantitative real-time PCR

Total RNA from tissue or sorted cells was isolated via Trizol (Invitrogen) or RNeasy kit (Qiagen). Total RNA from each sample was reverse transcribed with a first-strand cDNA synthesis kit (Quantabio or Thermo Fisher) according to the manufacturer's instructions. Quantitative real-time PCR reactions were performed with different sets of primers and Sensifast SYBR (Bioline, Taunton, MA) on a CFX96 or CFX384 Real-Time PCR Detection System (Bio-Rad; Bio-Rad CFX software 2.0) or Applied Biosystems QuantStudio 5 using 20 ng cDNA/reaction. Each quantitative real-time PCR reaction was performed at least twice and representative results are shown relative to a control by the standard $2^{-\Delta\Delta Ct}$ method, where Ct represents the number of cycles required to reach threshold for the target gene subtracted from the number of cycles required to reach threshold for a control housekeeping gene (*GAPDH* or *ACTB*). All data are shown relative to the first control condition in the given figure panel. Primer sequences are in Supplementary Data 2.

## SDS-PAGE and immunoblotting

Cells or tissues were homogenized with cold lysis buffer (RIPA buffer: 50 mM Tris-HCl pH 7.4, 150 mM NaCl, 0.1% SDS, 0.5% DOC, and 1% Nonidet P-40; coIP buffer: 20 mM Tris-HCl pH 8, 137 mM NaCl, 2 mM EDTA, 1% Nonidet P-40) supplemented with protease and phosphatase inhibitor cocktail (Thermo Fisher Scientific). Lysates (20 μg except where otherwise noted) were clarified by centrifugation at 14,000 × *g* for 15 min at 4 °C, electrophoresed under reducing conditions on 7.5%

or 4–20% gradient precast gels (Bio-Rad), and transferred to PVDF membranes (Bio-Rad and GVS Filter Technology). Membranes were blocked for 30 min in 5% Bovine Serum Albumin and incubated with primary antibodies against MRC2 (1:1,200: anti-mouse: AF4789, 1 μg/mL: anti-human: AF5770; R&D Systems, Bio-Techne), SEL1L (1:2,000; Novus Biologicals: NBP2-93746; ABCAM: ab78298), collagen I (1:2,000; Southern Biotech: 1310-01), human procollagen I alpha 1 (1:1,500; Novus Biologicals: AF6220), MYC (1:5,000; Cell Signaling Technology, 9B11), OS9 (1:2,000; Novus Biologicals: NB100-519), GAPDH (1:4,000; Cell Signaling Technology, 14C10), CD147 (1:2,000; Proteintech, 11989-1-AP), SHH (1:2,000; Cell Signaling Technology, C9C5), IRE1α (1:2,000; Cell Signaling Technology, 14C10), Vinculin (1:5,000; Cell Signaling Technology, E1E9V), or beta-Actin (1:10,000; Cell Signaling Technology, 13E5) overnight at 4 °C. Membranes were then incubated with horseradish peroxidase–conjugated secondary antibodies (anti-Rabbit, 1:4000: Cell Signaling Technology, 7074; anti-Sheep, 1:2,000: R&D Systems, Bio-Techne, HAF016; anti-Goat, Santa Cruz Biotechnology, sc-2354) for 1 h at room temperature. Blots were developed using enhanced chemiluminescence (Cytiva) by exposure onto film or ChemiDoc XRS+ (Bio-Rad; Image Lab 5.0). Where indicated densitometry was performed using ImageJ 1.53c (NIH), normalizing the protein of interest to a loading control (GAPDH or beta-Actin), and comparisons were made to the baseline condition. For immunoblots of FN2-deleted SEL1L, the Abcam antibody was used because the immunogen is distal to the FN2 domain, whereas the immunogen for the Novus antibody is the FN2 domain itself. Coomassie Brilliant Blue staining (Bio-Rad) was done according to the manufacturer's instructions after SDS-PAGE.

## Cell culture and molecular biology experiments

U937, MRC5 and HEK293T cells were obtained from ATCC (CRL-1593.2, CCL-171, CRL-3216, respectively). HEK293T cells that were WT, or null for SEL1L or HRD1 were obtained from Dr. Qi and were described previously[35,71–73]. HEK293T cells that are knock-in for SEL1LΔFN2 were created in the laboratory of Dr. Qi via CRISPR-Cas9 editing as previously described[74]. Briefly, WT HEK293T cells were electroporated with Cas9 and gRNAs (gRNA1 and gRNA3; Supplementary Data 2) flanking the SEL1L exon 4 which encodes SEL1L FN2 domain, and an additional gRNA (gRNA2; Supplementary Data 2) in close proximity to gRNA1 to enhance editing efficiency. These gRNAs were synthesized by Integrated DNA Technologies (IDT). These gRNAs and Cas9 protein were introduced into the cells via electroporation, followed by culturing and single-cell isolation with the desired genomic modification confirmed via Western blotting and sequencing. These cells have been characterized concurrently in Dr. Qi's laboratory and are described in detail in a recently accepted manuscript[51]. Immortalized MEFs from WT (*Sel1L^{f/f}*;*ERcre*-) or littermate inducible KO KO (*Sel1L^{f/f}*;*ERcre*+) mice were provided by Dr. Qi and were described previously[35]. MEFs were used at passage 4-16 and for experiments in which Sel1L deletion was induced, MEFs were used within 1 passage after addition of 4-hydroxytamoxifen (at 400 nM) vs. ethanol vehicle control as described previously[35]. Cells were grown in DMEM or RPMI (Corning) with 10% FBS (Gemini Bio-Products). ORFs containing full length murine SEL1L or GAPDH were obtained from Origene and were maintained in pCMV6 expression vectors and included an in-frame C-terminal tag. FN2 deletion was accomplished using Q5 Site-Directed Mutagenesis kit (New England Biolabs) according to the manufacturer's directions and in-frame deletion was verified with Sanger sequencing. ORFs (including the MYC tag) were also subcloned into pHIV lentiviral transfer plasmids using standard techniques and lentiviral particles were packaged in HEK293T cells with pMD2G and psPAX2. For MRC2 overexpression, pCI-Mrc2 (murine Mrc2 cloned into the pCI-Neo expression vector, from Promega) was used as we previously described[18]. All transfections were done using Lipofectamine 2000 (Invitrogen) according to the manufacturer's instructions.

For shRNA experiments, lentiviral particles harboring shRNA machinery within pSicoR-puromycin vectors were obtained from Sigma-Aldrich (scramble control: SHC002V; shRNA sequences can be found in Supplementary Data 2). Lentiviral infection was done in these experiments at 10 multiplicity of infection (MOI) in cells overnight in complete medium after addition of polybrene (8 µg/mL), and medium was replaced with fresh complete medium the next day. The infected cells were allowed to grow and then selected by resistance to puromycin (1 µg/mL, Sigma-Aldrich). Puromycin-selected cells were collected for experiments as indicated. Cells were infected at passages <10, and cells were collected for experiments between passages 5 and 16. For the luciferase assay, a 625 bp fragment around the MRC2 transcriptional start site in a Firefly luciferase vector with a minimal promoter element (Promega) was used as previously described[18]. The Stop&Glo® (Promega) system was used to read luciferase activity in cells on a luminometer (Synergy H1 Plate Reader Gen5 3.02). All firefly luciferase values were normalized to luciferase levels of a constitutive Renilla for a transfection control. Cycloheximide (20 µg/mL, Sigma-Aldrich) and MG-132 (10 µM, EMD Millipore) were used for time periods indicated in the text or figures and always compared to vehicle controls (DMSO).

## Co-immunoprecipitation

Proteins were isolated in non-denaturing coIP lysis buffer (see above). Proteins were incubated with fast-flow protein G- or protein A- Dynabeads (Invitrogen) for 2 h and centrifuged to eliminate nonspecifically bound proteins. The concentrations of the precleared proteins were estimated by Bradford's assay. Primary antibodies against Procollagen (Novus) or SEL1L (Novus) or a non-specific IgG control antibody were incubated with protein G- or protein A-Dynabeads for 4 h. Then, 200–400 µg pre-cleared protein were incubated overnight with the antibody-bound beads. After washing, attached proteins were then eluted from the beads in 1% SDS buffer by heating at 60 °C and vortexing followed by centrifugation to collect the supernatants as the eluates. Equal volumes of the eluates were used for immunoblotting. Immunoprecipitation with protein isolation buffer without lysates was run as a negative control for the beads. Proteins immunoprecipitated using non-specific IgG (NS IgG) were negative control for specific antibodies. Immunoblotting with total pre-cleared protein lysates were used as input samples. For purification of the WT and ΔFN2 proteins, MYC-tagged protein was purified from transduced cells (see above) with Myc-trap agarose (Chromotek), a nanobody based system, according to the manufacturer's instructions; these constructs also contained a C-terminal FLAG tag and other purification studies were done using anti-FLAG magnetic agarose (Thermo Scientific Pierce) according to the manufacturer's instructions. For co-IP experiments with purified collagen, purified WT or ΔFN2 proteins were incubated with rat tail collagen (EMD Millipore) and then co-IP was carried out as described above.

## Circular dichroism

WT or ΔFN2 purified protein samples were diluted to final concentration of ~0.35 mg/mL in a Tris-Glycine buffer. CD spectra were taken with Chirascan V100 CD Spectrometer (Applied Photophysics) using a 0.5-mm quartz cuvette between 200 and 300 nm at a set temperature of 25 °C.

## Flow cytometry and flow cytometric-based collagen uptake assay

Flow cytometric analysis was done using a FACSVerse, FACSAriaII, or Accuri c6 (BD Biosciences) and sorting was done using FACSAriaII devices (BD Biosciences) at the UCSF Parnassus Flow Cytometry Core; acquisition software used was FACSDiva 6 or higher, or Accuri c6 software v1.0. For the collagen uptake assay, cells were incubated in fresh media with Oregon Green 488-conjugated gelatin (10 µg/ml final concentration; Life Technologies) for 1 h at 37 °C. Cells were then harvested (for suspension cells) or trypsinized and harvested for analysis. To assess only internalized fluorescent collagen (i.e. to ignore bound but not internalized collagen), cells were either washed before flow cytometry or Trypan blue was added to quench bound but not internalized fluorochromes as we have done previously[21,64]. For cell-surface staining, cells were stained using a standard FACS staining protocol. Antibodies included: anti-MRC2 (as above) as a primary antibody with a secondary of Donkey anti-sheep conjugated to a fluorophore (1:250, Invitrogen, A-21448). Cells were then analyzed and/or sorted on one of the above devices, with sorting and analysis gates based on fluorescence-minus-one controls after compensation was performed with single-stain controls. Data were analyzed with FlowJo, version 7.6.1 (Tree Star, Ashland, OR).

## Immunofluorescence imaging

Cells plated on coverslips were fixed with 4% PFA in PBS. Coverslips were blocked in Phosphate-buffered saline (PBS) with 0.5% Bovine serum albumin, 0.1% Triton X-100 and 5% donkey serum. Antibodies against Type I collagen (Southern Biotech, as above or Rockland, 600-401-103), MRC2 (R&D, as above), Calnexin (Cell Signaling Technology, C5C9), MYC (Santa Cruz Biotechnology, 9E10), or SEL1L (Novus, as above for Fig. 7a and Supplementary Fig. 4b; for Fig. 7d, anti-SEL1L was provided as a gift by Dr. Qi[35,71–73] and was used since it can at least partially recognize SEL1L without its FN2 domain because the immunogen includes peptide sequence outside the FN2 domain) were applied at 1:100 dilution followed by secondary antibody conjugated to Alexa Fluor 488 or 594 or 647 (Invitrogen, A-1105, A-21207, A-21448, A-31571 or A-11015) at 1:100. Coverslips were mounted in Vectashield (Vector Laboratories). Human lung fragments were fixed in 4% PFA in PBS and then dehydrated in sucrose and embedded in OCT and frozen in chilled isopropanol on dry ice. Frozen sections (30 µm) were prepared. Sections were blocked in Phosphate-buffered saline (PBS) with 0.5% Bovine serum albumin, 0.1% Triton X-100 and 5% donkey serum followed by staining with primary and secondary antibodies as described above. Sections were mounted in Vectashield (Vector Laboratories).

Confocal images were captured using a Leica TCS SPE confocal microscope with an ACS APO 40× or 63× Oil CS objective lens at room temperature with Leica Type F Immersion Liquid (11513859) or with a Leica Stellaris 8 confocal microscope with a 10× air or 40× Oil objective (Leica LASX 4.4). Wide-field epifluorescence images were captured by an Olympus DP70 CCD using a standard Olympus BX51 upright microscope with an Olympus UplanFL 40x objective at room temperature or with a Nikon TE2000-U microscope and Ds-Qi2 camera (Nikon NIS-Elements 5.20). Images were processed with ImageJ 1.53c (National Institutes of Health, Bethesda, Maryland, USA). All imaging was done on the same platform within any given experiment. Images were processed with ImageJ, in which any brightness/contrast was changed uniformly across all conditions and pseudocoloring was applied. Single optical slices at Airy 1 are shown for confocal images.

## Human lung tissue

Normal human lung tissue from deceased donors of different ages was obtained from lungs not used by the Northern California Transplant Donor Network according to an IRB-exempted protocol led by Dr. Wolters; IPF lung specimens were obtained from explanted lungs removed during lung transplantation, in an IRB-approved protocol led by Dr. Wolters (see Supplementary Data 4 for more details). After harvest, lung tissue was directly snap-frozen in liquid nitrogen and protein was extracted as above or 1 cm fragments were fixed in 4% PFA in PBS and processed as described above for imaging.

## In silico protein structure prediction

Docking of the FN2 domain of SEL1L (Uniprot Q9UBV2; AlphaFoldDB structural prediction) with a collagen peptide sequence

(PGASGPMGPRGPPGPPGKNG) was done using MDockPeP[75,76] and visualized with PyMOL (version 2.3; Schrödinger). For prediction of the structures of full length SEL1L and the ΔFN2 mutant, the Robetta online platform (which relies on RoseTTAFold prediction algorithm[77]) was used to predict the structure of both the full-length sequence (Uniprot Q9Z2G6) and the sequence without AA118-166 (the FN2 domain). Alignment and visualization were performed in PyMOL 2.3.

## Proteomics

Gel bands were excised after Coomassie Brilliant Blue staining. Then in-gel trypsin digestion was performed, followed by stage-tip desalting and LC-MS/MS. The data were processed by MaxQuant v2. MS data was searched against Uniprot human protein database with the addition of the target sequences since these come from mouse SEL1L (note that in the database, the WT mouse SEL1L has an accession of O00001; the in-frame ΔFN2 mutant, O00002).

## Statistics

Data were evaluated with Graphpad Prism 9 software (San Diego, CA): for two sample comparisons, by 2-tailed Student's *t* test or Mann–Whitney *U* test if data did not pass a test of normality; for multiple comparisons, by ANOVA with post-hoc Bonferroni, Tukey, or Dunnett multiple comparison testing (as suggested by Prism statistical analysis software) if the ANOVA was significant; by ANCOVA; or by Pearson correlation as indicated. A *p*-value of 0.05 or less was considered statistically significant with *=$p < 0.05$, **=$p < 0.01$, ***=$p < 0.001$, ****=$p < 0.0001$, NS=Not Significant. Data are presented as mean ± SEM.

## Reporting summary

Further information on research design is available in the Nature Portfolio Reporting Summary linked to this article.

## Data availability

Source data are provided with this paper. All data generated or analyzed during this study are included in this published article (and its Supplementary items). Proteomic data have also been deposited in PRIDE with accession PXD048563. Prior data from the Gene Expression Omnibus (GEO) used for re-analysis can be found on the GEO database: GSE136831, GSE135893, GSE121611, GSE128033, GSE132771, GSE110147, GSE70867, GSE37858, GSE40151. Source data are provided with this paper.

## Code availability

The codes used in CRISPR screen analysis were previously published and are described with appropriate citations above.

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

## Acknowledgements

This research was supported by US National Institutes of Health grants K08 HL145015 (M.J.P.), R01 HL136377 (K.A.), R01DK132786 (L.Q.), R35GM130292 (L.Q.), a Grant-in-aid from The Stony Wold-Herbert Fund (M.J.P.), and a Sponsored Research Agreement with Celgene (K.A.). We acknowledge the UCSF Parnassus Flow CoLab (RRID:SCR_018206) for assistant generating flow cytometry data, supported in part by Grant NIH P30 DK063720 and by the NIH S10 Instrumentation Grant S10 1S10OD021822-01. We acknowledge and thank the Precision Biomolecular Characterization Facility at Columbia University for technical support and access to the Circular Dichroism spectrometer, which is supported by NIH Award 1S10OD025102-01. Mass spectrometric analysis was performed at the Weill Cornell Medicine Proteomics and Metabolomics Core Facility. We would like to thank S. Layer for ongoing inspiration.

## Author contributions

MJP and KA conceived the study, supervised the study, designed the experiments, and wrote and revised the manuscript. MJP carried out the experiments, conducted the analysis, and generated the figures, with the assistance of BK, MP, AHB, and CDY. Additional intellectual and analytical input was provided by COL, RD, PJW, MTM, and LQ. Key reagents were provided by LLL, ZW and LQ. All authors approved the final paper.

## Competing interests

The authors declare no competing interests.
