## [Peer Review File · Nature Communications]

REVIEWER COMMENTS

Reviewer #1 (Remarks to the Author):

In this paper, the authors use unbiased CRISPR-based high-throughput screens, followed by focused knock-down/knock-out strategies, to identify pathways responsible for cellular collagen clearance in normal homeostasis and fibrosis. They interpret their results to delineate a mechanism in which collagen expression / secretion, in its own right, is monitored by the ER protein, SEL1L, which in turn leads to upregulation of the endocytic collagen receptor uPARAP/Endo180 (product of the MRC2 gene and referred to as MRC2 in the paper). This upregulation leads to collagen uptake capacity through this receptor. The authors then suggest that a de-coupling of this mechanism may occur in non-resolving fibrotic conditions such as IPF. If fully proven, the proposed mechanism would represent a substantial contribution to knowledge in the field.

Altogether, the paper includes a number of well substantiated observations, giving good support to some of the partial conclusions stated. However, to demonstrate the complete mechanism as summarized above, a considerably more thorough documentation of several steps would be needed. A more detailed description follows.

Using cellular uptake of fluorescent collagen (i.e. gelatin) as the read-out, the authors start out to perform both CRISPRi- and CRISPRa-based screens, designed to cover all genes, in U937 cells. This investigation very strongly pinpoints the MRC2 gene product as a central player in collagen uptake, being identified with striking clarity in both the CRISPRi and the CRISPRa screen. Although this finding is not new, the fact that this receptor occurs as a very unique component in both of the unbiased global screens makes this observation valuable in its own right.

The same screens point to additional relevant components with varying degrees of certainty, in most cases being evident in only one of the screens (CRISPRi or CRISPRa). Some of these latter components form the basis for the subsequent studies. Thus, using bioinformatics (Gene Ontology) studies, the authors observe a pronounced occurrence of genes related to collagen metabolism, including collagen production, among hits from the screens. Although it is difficult to judge how striking this preference actually is, the authors succeed to demonstrate the importance of one such hit, TRAM2, with importance for collagen biosynthesis. They show that silencing of TRAM2 using shRNA leads to both a decrease in collagen production, MRC2 gene expression and cellular uptake of collagen.

In the next part of the paper, based on the hypothesis that cells might possess a sensing mechanism for collagen production, the authors focus on an additional hit from the CRISPRi screen, SEL1L, and perform a number of studies leading to the composite model referred to above. These latter studies raise various concerns summarized in the points below, which also include certain reservations regarding observations mentioned already.

Major points:

- 1) For several western blot (WB) experiments in the study, the interpretation of up- or down regulation appears somewhat uncertain. Although loading controls are indeed included in all cases, the amounts of the chosen housekeeping proteins, etc, are subject to inevitable variations, as also found here.

Unfortunately, this makes it difficult to judge with certainty about the regulation of the focus components analyzed on the same gels. This is the case in Figs. 3F, 4C, 5B and left part of 5D.

2) Experiments with shRNA-based knock-down of collagen expression (Fig. 3I-K) are quite important in the line of evidence leading to the authors suggesting the over-all mechanism brought forward in the paper. These experiments were done in U937 (leukemic) cells. Although the authors present both published studies and actual analysis to verify that such cells could have some relevance in collagen production, leukocyte-derived cells are not generally considered major players in this connection. Therefore, it is doubtful whether mechanisms found in those cells can be safely generalized to other cell types. For the same reason, the authors also include other studies using MRC5 fibroblasts instead of U937 cells. However, the latter studies are based on relatively unspecific stimulation of collagen production, making the direct connection with the processes of interest less certain.

3) The experiments addressing SEL1L are based on an inducible knockout mouse embryonic fibroblast cell line. While this may indeed be an excellent approach, there is much too little description of these cells and the control cells. This information is also lacking in the Methods section. Although a reference to these cells is provided, this is insufficient to truly document that the target gene is the only variable in play when comparing the knockout and the control cells. Clonal variations, occurring gradually upon long term culture of fibroblasts, are not uncommon. Were the knockout and control cells derived from littermate mice, and how many passages were used when growing the cells? These concerns are particularly important because some of the effects observed are quite modest and/or subject to a large variation.

4) Although it is an attractive hypothesis that the FN2 domain of SEL1L might interact with collagen, such a hypothesis cannot be verified by the use of a domain deletion mutant (Fig. 6). It is well established that deletion mutants may affect other protein regions, over-all protein folding, etc. Furthermore, the co-IP and immunofluorescence experiments presented are not sufficient to demonstrate this kind of interaction, which would require a more detailed molecular analysis.

5) In several cases, there is some tendency to exaggerate the conclusions that can be drawn from a limited number of observations. Examples include:

“This informatic analysis suggested that collagen biosynthetic activity positively regulates cellular uptake of extracellular collagen fragments” (P. 5, line 116-117) and “These data indicate that there is a biological basis for positive regulation collagen uptake by collagen biosynthesis genes, and this is specific to uptake of collagen. (P. 5, line 122-123) (In both cases, this is hardly possible to conclude based on the CRISPr screens alone);

“These data together suggest a conserved pathway in which collagen biosynthesis directly regulates MRC2 expression and collagen uptake” (P. 6, line 151-152) (A very comprehensive hypothesis, based on just a few knock-down experiments or relatively unspecific stimulation studies);

“... the FN2 domain is only present in vertebrates and higher organisms, showing the collagen-sensing function arose later and is likely to be independent of its original ERAD function” (P. 9, line 206-208) (This is insufficient to serve as an argument in support of the role of the FN2 domain in this connection);

“Given that we have previously shown that MRC2 is necessary for resolution of fibrosis¹³, these data indicate that the impairment of biosynthesis-induced collagen clearance via MRC2 could be an important

pathway driving non-resolving fibrosis and IPF." (P. 9, line 225-227) (Appears too far-fledged when based on just the correlative studies shown in Fig. 7).

Minor points:

Comparison of enrichment of the Gene Ontology terms in the collagen uptake screens with data from previously published CRISPR screens of phagocytosis of other cargo (P. 5, line 117-119): This comparison is hardly relevant. The cellular collagen uptake through the MRC2 gene product is not a phagocytic process; see Sprangers et al. 2017 (PMID: 27922193).

Fig. 3A: It should be made more clear in the legend how the presented top genes from the Gene Ontology gene set were selected.

Fig. 4, legend: Presumed typo. It must be WT = Sel1L^{f/f};ERcre⁻, iKO = Sel1L^{f/f};ERcre⁺; not vice versa.

Fig. 6A: The fluorescence in both channels is so strong that it is hard to judge about co-localization.

Fig. 7: Presumed typo (N numbers). It appears from the figure that the larger numbers refer to IPF and the smaller numbers to normal.

Reviewer #2 (Remarks to the Author):

The manuscript by Podolsky et al. describes the novel SEL1L role as an intracellular rheostat that controls collagen turnover. In mammals, collagen is the most abundant component and is constantly synthesized and degraded to maintain the amount. Despite the imbalances of collagen turnover leading to fibrotic diseases such as IPF, little was known about the mechanisms that regulate collagen clearance. In this study, the authors identified SEL1L involves collagen clearance by using a genome-wide screening. They found that SEL1L senses the amount of collagen synthesis through its FN2 domain and regulates the extracellular collagen uptake by regulating the MRC2 expression. However, their data are insufficient to prove that SEL1L regulates collagen turnover as an intracellular rheostat. The major concerns are listed below.

1. In this paper, the binding between SEL1L and collagen provides essential evidence that SEL1L directly senses collagen. However, despite the overexpression of SEL1L, the binding is too low. This result cannot deny the possibility that collagen indirectly binds to SEL1L through other factors. In addition, under the Δ FN2 condition, the collagen expression level itself was changed.

Thus, it is difficult to determine that FN2 is the binding site.

2. The authors concluded that the sensing of collagen content by SEL1L regulates the MRC2 expression. However, how collagen sensing by SEL1L influences the regulation of MRC2 expression levels is entirely unknown. This paper only showed a correlation between the SEL1L expression level and MRC2. Therefore, there is little evidence to support the claim that SEL1L regulates MRC2 expression or collagen turnover.

3. The authors concluded that the collagen clearance by SEL1L is a completely independent system from ERAD. In Figure 5, the authors showed that HRD1 deficiency did not regulate MRC2 stability. However, this is indirect evidence, and there is no evidence that ERAD inhibition by SEL1L deficiency is completely

the same as HRD1 deficiency. Without knowing how collagen sensing by SEL1L regulates MRC2, it is difficult to rule out the effects of ERAD completely. Moreover, considering the previous report (DOI:<https://doi.org/10.1074/jbc.M110.215871>) showing HRD1 contributes to SEL1L stability, HRD1 deficiency might affect the SEL1L expression. Indeed, HRD1 deficiency reduced the expression of SEL1L in FIG. 5D.

4. The discussion of IPF-causing mutations is unclear, especially since the causal relationship between this mechanism and IPF is not fully explained.

5. Overall, descriptions such as figure numbers and term descriptions are insufficient.

Reviewer #3 (Remarks to the Author):

The manuscript by Podolsky et al used genome-wide screening to identify the role of SEL1L in the context of collagen metabolism. This investigation aims to improve the understanding on the uptake and degradation of collagens in the context of fibrotic diseases, especially idiopathic pulmonary fibrosis. The data supports the hypothesis of the authors; however, the data presentation needs to be improved.

Major criticisms:

1. The introduction is based on four publications published before the year 2000 (Ref# 4, 6-8). There are a number of newer publications on this topic, which can easily be found by Pubmed searches, e.g. Inoue et al. *Int J Mol Sci* 2023. pp6695; Costanzo et al. *Pulm Med* 2022. pp3632764; Mahalanobisch et al. *Pharmacol Res* 2020. pp104591. etc.

2. Introduction, line 67: References 1 and 3 are not relevant citations. Line 68: the authors need to provide more details on the mentioned novel mediators in the two citations. Line 73: it is not clear what the authors meant by “genetic regulatory pathways”? Do they mean gene regulation? Can such genetic modification really explain deregulated collagen processing in fibrosis? More relevant to this question would be if the encoded enzymes are activated or if they are inhibited, which can both occur on the epigenetic level. Lines 75-79: It is unusual that the authors describe their findings in their introduction.

3. The methods lack a detailed description and number of the control and the IPF tissue donors. It is not clear to this reviewer why the authors used U937 cells, which are monocytes and not macrophages that would be more relevant to study collagen metabolism/degradation. Line 397: MRC5 cells are fibroblast-based, and there is no explanation in which experiments they have been used. There is also a question which role fibroblasts play in the context of collagen metabolism and degradation. Line 422: It is mentioned that human lung fibroblasts were isolated from tissues, but there is no information where they have been used in this study, nor in which medium and condition they were expanded and characterised. Line 397: Why were HEK293T cells (embryonal kidney epithelial cells) used for the expression of the plasmids? How is this cell type relevant to IPF.

4. The discussion is very short and does not sufficiently explain how the data is linked to collagen uptake and degradation in fibrotic lung diseases, specially IPF. Unfortunately, the authors do not include the discussion on the role of MRC2 in the context of fibrotic diseases. MRC2 is a mannose receptor, which contributes to fibroblast proliferation by inhibiting apoptosis (Li et al. *Analytical Cellular Pathol* 2021. pp6619870; Onursal et al. *Frontiers Med* 2021.).

5. There is also no description on the study limitations.

6. Figures: In supplementary figure 2, it is not clear why these marked pathways were selected? On which data basis was the selective condition of the “collagen-related genes” defined? Figure 3 is too crowded and needs to be simplified or split. Figure 3F is not necessary for understanding collagen processing. It can be added as supplementary information. The representative Western-blot shown for figure 3L-P should be presented as supplementary information. Figures 4C, 4H, 5D, 6C, 6E should be provided as supplementary figures. Instead, the authors should demonstrate more clearly the effect of the investigated silencing and gene deletions on collagen uptake.

Reviewer #4 (Remarks to the Author):

In their study, Podolsky et al, performed unbiased genome-wide screens to understand the molecular mechanisms of cell-mediated collagen clearance. They report that a sensing mechanism that is dependent on SEL 1L protein and occurs via a noncanonical function of SEL 1L regulates clearance of extracellular collagen and this may be of great significance for fibrotic lung diseases.

The study is very interesting; however, it needs to address the following points:

1. There is no information regarding the donors of IPF and of control lungs. All information regarding demographics and clinical data of donors should be included in a Table.
2. How many IPF and control lung tissues were used per experiment? In Figure 8 it is indicated that N=7 for the western blot but it is not clear if this is also the case for the immunofluorescence and for the PCR experiments.
3. There is no description of the procedure used to isolate and establish primary cultures of human lung fibroblasts. From how many donors was this performed? Why the authors did not grow primary fibroblasts from IPF lung tissues as well?
4. The authors should also provide the reasons why they have used the type of cells that they used in their study.
5. The discussion of the manuscript is very short. The authors should put their results in context to IPF and other fibrotic diseases that may share similar pathophysiological mechanisms and explain how the results presented in this manuscript can be used in a clinical setting. The limitations of the study should also be discussed in a paragraph of the discussion.

Response to reviews

We thank the editor and reviewers for their thoughtful comments and suggestions. We have attempted to address all comments and concerns both with new experiments and in the text of the manuscript and believe these changes have significantly strengthened the manuscript. Specifically, here is a point-by-point response to the reviews:

Reviewer 1:

Major points:

- 1- *“For several western blot (WB) experiments in the study, the interpretation of up- or down regulation appears somewhat uncertain. Although loading controls are indeed included in all cases, the amounts of the chosen housekeeping proteins, etc, are subject to inevitable variations, as also found here. Unfortunately, this makes it difficult to judge with certainty about the regulation of the focus components analyzed on the same gels. This is the case in Figs. 3F, 4C, 5B and left part of 5D.”*

Response: We have added densitometry of the multiple blots referenced for each experiment to clarify the findings.

- 2- *“Experiments with shRNA-based knock-down of collagen expression (Fig. 3I-K) are quite important in the line of evidence leading to the authors suggesting the over-all mechanism brought forward in the paper. These experiments were done in U937 (leukemic) cells. Although the authors present both published studies and actual analysis to verify that such cells could have some relevance in collagen production, leukocyte-derived cells are not generally considered major players in this connection. Therefore, it is doubtful whether mechanisms found in those cells can be safely generalized to other cell types. For the same reason, the authors also include other studies using MRC5 fibroblasts instead of U937 cells. However, the latter studies are based on relatively unspecific stimulation of collagen production, making the direct connection with the processes of interest less certain.”*

Response: To address this concern, we have directly silenced COL1A1 in MRC5 fibroblasts and show that this leads to a reduction in collagen uptake and MRC2 protein, as expected based on our findings in U937 cells (Fig. 3L-N)

- 3- *“The experiments addressing SEL1L are based on an inducible knockout mouse embryonic fibroblast cell line. While this may indeed be an excellent approach, there is much too little description of these cells and the control cells. This information is also lacking in the Methods section. Although a reference to these cells is provided, this is insufficient to truly document that the target gene is the only variable in play when comparing the knockout and the control cells. Clonal variations, occurring gradually upon long term culture of fibroblasts, are not uncommon. Were the knockout and control cells derived from littermate mice, and how many passages were used when growing the cells? These concerns are particularly important because some of the effects observed are quite modest and/or subject to a large variation.”*

Response: We have added details on this in the Methods section.

- 4- *“Although it is an attractive hypothesis that the FN2 domain of SEL1L might interact with collagen, such a hypothesis cannot be verified by the use of a domain deletion mutant (Fig. 6). It is well established that deletion mutants may affect other protein regions, over-all protein folding, etc. Furthermore, the co-IP and immunofluorescence experiments presented are not sufficient to demonstrate this kind of interaction, which would require a more detailed molecular analysis.”*

Response: We have added multiple new pieces of data to address this concern. We show computational prediction of docking of collagen peptide into the FN2 domain (Supp. Fig. 4) and new additional experimental evidence of interaction based on co-IP of collagen and purified WT vs. FN2 mutant (Supp. Fig. 5). To address concerns about the folding and stability of the mutant, we show computational prediction of nearly complete overlap between the full length and mutant (Supp. Fig. 5), showing no change in remaining protein structure is expected based on the RoseTTAFold engine. In addition, we show that the mutant construct is translated and is of the expected molecular weight and that the full length and mutant proteins both co-localize with the ER marker Calnexin, showing that the mutant is in the right subcellular localization (Supp. Fig. 5). Finally, we present circular dichroism spectra of the purified full length and mutant proteins that are nearly overlapping, suggesting no changes in secondary structure caused by the mutant construct (Supp. Fig. 5).

- 5- *"In several cases, there is some tendency to exaggerate the conclusions that can be drawn from a limited number of observations. Examples include:
"This informatic analysis suggested that collagen biosynthetic activity positively regulates cellular uptake of extracellular collagen fragments" (P. 5, line 116-117) and "These data indicate that there is a biological basis for positive regulation collagen uptake by collagen biosynthesis genes, and this is specific to uptake of collagen. (P. 5, line 122-123) (In both cases, this is hardly possible to conclude based on the CRISPr screens alone);
"These data together suggest a conserved pathway in which collagen biosynthesis directly regulates MRC2 expression and collagen uptake" (P. 6, line 151-152) (A very comprehensive hypothesis, based on just a few knock-down experiments or relatively unspecific stimulation studies);
"... the FN2 domain is only present in vertebrates and higher organisms, showing the collagen-sensing function arose later and is likely to be independent of its original ERAD function" (P. 9, line 206-208) (This is insufficient to serve as an argument in support of the role of the FN2 domain in this connection);
"Given that we have previously shown that MRC2 is necessary for resolution of fibrosis13, these data indicate that the impairment of biosynthesis-induced collagen clearance via MRC2 could be an important pathway driving non-resolving fibrosis and IPF." (P. 9, line 225-227) (Appears too far-fledged when based on just the correlative studies shown in Fig. 7)."*

Response: We have updated the language used in the manuscript as suggested.

Minor points:

- 1- *"Comparison of enrichment of the Gene Ontology terms in the collagen uptake screens with data from previously published CRISPR screens of phagocytosis of other cargo (P. 5, line 117-119): This comparison is hardly relevant. The cellular collagen uptake through the MRC2 gene product is not a phagocytic process; see Sprangers et al. 2017 (PMID: 27922193)."*

Response: We appreciate that MRC2-mediated collagen uptake is not a phagocytic process, but some genes identified as hits in our uptake screen were also identified as hits in the referenced cargo uptake screens (e.g. *RAB1A* and *INPP5D*). We therefore believe the data in this figure showing specificity of the highlighted pathways to the collagen uptake screen has some relevance (our screen was a phenotypic screen for fluorescent collagen uptake, which at least in theory could be taken up by phagocytic pathways as well as non-phagocytic pathways).

- 2- *"Fig. 3A: It should be made more clear in the legend how the presented top genes from the Gene Ontology gene set were selected."*

Response: We have clarified how the genes from 3A were selected in the legend as suggested.

- 3- *"Fig. 4, legend: Presumed typo. It must be WT = Sel1Lf/f;ERcre-, iKO = Sel1Lf/f;ERcre+; not vice versa."*
Response: Yes this was a typo and it has been corrected -thank you for pointing this out.
- 4- *"Fig. 6A: The fluorescence in both channels is so strong that it is hard to judge about co-localization."*
Response: We have added additional images with lower brightness (Supp. Fig. 4B).
- 5- *"Fig. 7: Presumed typo (N numbers). It appears from the figure that the larger numbers refer to IPF and the smaller numbers to normal."*
Response: We have corrected this typo - thank you for pointing this out.

Reviewer 2:

- 1- *"In this paper, the binding between SEL1L and collagen provides essential evidence that SEL1L directly senses collagen. However, despite the overexpression of SEL1L, the binding is too low. This result cannot deny the possibility that collagen indirectly binds to SEL1L through other factors. In addition, under the Δ FN2 condition, the collagen expression level itself was changed. Thus, it is difficult to determine that FN2 is the binding site."*
Response: See our response to reviewer 1, point 4 above which uses multiple methods to demonstrate the binding.
- 2- *"The authors concluded that the sensing of collagen content by SEL1L regulates the MRC2 expression. However, how collagen sensing by SEL1L influences the regulation of MRC2 expression levels is entirely unknown. This paper only showed a correlation between the SEL1L expression level and MRC2. Therefore, there is little evidence to support the claim that SEL1L regulates MRC2 expression or collagen turnover."*
Response: We show in this manuscript that SEL1L positively regulates MRC2 via loss-of-function and gain-of-function approaches (Fig. 5 and 6). We go on to show, with our original data and the data we have added to the paper, that SEL1L-dependent regulation of MRC2 requires the FN2 domain which is responsible for collagen binding, which we have now characterized in greater detail. Furthermore, we demonstrate this occurs at the mRNA level (Fig. 5-7) and we show that this is likely occurring at the gene transcript production level according to luciferase data (Fig. 6D). Taken together, we believe these data show more than correlation but actual regulation of MRC2 by SEL1L that requires the FN2 domain; it is possible that the FN2 domain is necessary for regulating MRC2 via another mechanism than collagen binding, although our data in Fig. 3 supports the idea that collagen synthesis regulates MRC2, and we do acknowledge this in our Discussion section. However, additional work showing how the FN2 domain binding collagen leads to MRC2 transcription is beyond the scope of this manuscript and is an area of active research in our lab for a subsequent publication.
- 3- *"The authors concluded that the collagen clearance by SEL1L is a completely independent system from ERAD. In Figure 5, the authors showed that HRD1 deficiency did not regulate MRC2 stability. However, this is indirect evidence, and there is no evidence that ERAD inhibition by SEL1L deficiency is completely the same as HRD1 deficiency. Without knowing how collagen sensing by SEL1L regulates MRC2, it is difficult to rule out the effects of ERAD completely. Moreover, considering the previous report (DOI:<https://doi.org/10.1074/jbc.M110.215871>) showing HRD1 contributes to SEL1L stability, HRD1 deficiency might affect the SEL1L expression. Indeed, HRD1 deficiency reduced the expression of SEL1L in FIG. 5D."*
Response: We thank the reviewer for this suggestion. In addition to the data we previously included in the manuscript, we present new additional data showing that ERAD substrate OS9 accumulates in SEL1L knockout (as expected), but that either a

WT or FN2-deletion mutant will rescue this phenotype (Fig. 7). However, the FN2-mutant does not rescue the downregulation of MRC2. We believe this provides strong support for the hypothesis that SEL1L-mediated positive regulation of MRC2 is not dependent on ERAD (in addition to the fact that HRD1 does not positively regulate *MRC2* message). We cannot rule out that MRC2 protein could also be an ERAD substrate (see expanded Discussion section), but the mechanism by which SEL1L positively regulates *MRC2* message we have characterized does not require ERAD.

- 4- *“The discussion of IPF-causing mutations is unclear, especially since the causal relationship between this mechanism and IPF is not fully explained.”*

Response: We have expanded our discussion of the relationship between SEL1L and IPF in both Fig. 9 and in our Discussion section; we do not believe any known common genetic mutations that cause IPF are necessarily related to SEL1L.

- 5- *“Overall, descriptions such as figure numbers and term descriptions are insufficient.”*

Response: We have expanded descriptions and figure legends.

Reviewer 3

- 1- *“The introduction is based on four publications published before the year 2000 (Ref# 4, 6-8). There are a number of newer publications on this topic, which can easily be found by Pubmed searches, e.g. Inoue et al. Int J Mol Sci 2023.pp6695; Costanzo et al. Pulm Med 2022. pp3632764; Mahalanobis et al. Pharmacol Res 2020. pp104591. etc.”*

Response: We have expanded our introduction and thank the reviewer for the suggestions.

- 2- *“Introduction, line 67: References 1 and 3 are not relevant citations. Line 68: the authors need to provide more details on the mentioned novel mediators in the two citations. Line 73: it is not clear what the authors meant by “genetic regulatory pathways”? Do they mean gene regulation? Can such genetic modification really explain deregulated collagen processing in fibrosis? More relevant to this question would be if the encoded enzymes are activated or if they are inhibited, which can both occur on the epigenetic level. Lines 75-79: It is unusual that the authors describe their findings in their introduction.”*

Response: We have amended and edited the introduction as suggested and to clarify what we mean. We agree with the reviewer that upstream regulation of this process at the gene level may not be the only way in which collagen turnover is regulated or dysregulated in fibrosis and translational or post-translational effects are likely quite important; however, this does not rule out the additional importance of regulation at the gene or gene transcriptional level. Furthermore, the screens we employed are agnostic to whether the effect of gene interference mechanistically operates at a transcriptional or other level, since the readout is based on the collagen uptake phenotype, a functional outcome. The approach we took is also somewhat more feasible than genome-wide screens that manipulate post-translational (or epigenetic or other levels of regulation) effects of proteins on this outcome, though this would be worthy of future study and something we plan to pursue going forward.

- 3- *“The methods lack a detailed description and number of the control and the IPF tissue donors. It is not clear to this reviewer why the authors used U937 cells, which are monocytes and not macrophages that would be more relevant to study collagen metabolism/degradation. Line 397: MRC5 cells are fibroblast-based, and there is no explanation in which experiments they have been used. There is also a question which role fibroblasts play in the context of collagen metabolism and degradation. Line 422: It is mentioned that human lung fibroblasts were isolated from tissues, but there is no information where they have been used in this study, nor in which medium and condition*

they were expanded and characterised. Line 397: Why were HEK293T cells (embryonal kidney epithelial cells) used for the expression of the plasmids? How is this cell type relevant to IPF.”

Response: Use of U937 cells was done primarily for two reasons: (1) these cells readily take up collagen and express MRC2, a relevant collagen receptor that has proven *in vivo* importance and (2) these cells grow in suspension and can be genetically manipulated, allowing genome-wide screening with high coverage (i.e. culturing of >300-500 x 10⁶ cells in one bioreactor) – whereas macrophages and monocytes are not easily genetically manipulable for example and adherent cells significantly complicate cell culture of large numbers of cells – this rationale is detailed at the end of the first paragraph of the ‘Results’ section. We have now included a detailed description and table of our use of human lung tissue (see Supp. Table 3 and expanded Results and Methods sections). Additionally, the header of the Methods section referring to primarily isolated human lung fibroblasts was a typographical error - we did not include primary human lung fibroblasts in this manuscript and have amended the title of that section appropriately. We have expanded our discussion on the use of the different cell types in this manuscript at various points in the manuscript. Finally we have pointed out where MRC5 human lung fibroblasts (vs. any other cell type) were used in this manuscript – these were used for many validation studies since the original screen cell type was U937 cells.

- 4- “The discussion is very short and does not sufficiently explain how the data is linked to collagen uptake and degradation in fibrotic lung diseases, specially IPF. Unfortunately, the authors do not include the discussion on the role of MRC2 in the context of fibrotic diseases. MRC2 is a mannose receptor, which contributes to fibroblast proliferation by inhibiting apoptosis (Li et al. *Analytical Cellular Pathol* 2021. pp6619870; Onursal et al. *Frontiers Med* 2021.).”

Response: We have expanded the Discussion as above.

- 5- “There is also no description on the study limitations.”

Response: We have also included study limitations in the Discussion.

- 6- “Figures: In supplementary figure 2, it is not clear why these marked pathways were selected? On which data basis was the selective condition of the “collagen-related genes” defined? Figure 3 is too crowded and needs to be simplified or split. Figure 3F is not necessary for understanding collagen processing. It can be added as supplementary information. The representative Western-blot shown for figure 3L-P should be presented as supplementary information. Figures 4C, 4H, 5D, 6C, 6E should be provided as supplementary figures. Instead, the authors should demonstrate more clearly the effect of the investigated silencing and gene deletions on collagen uptake.”

Response: We have amended the figure legend to explain how the various pathways were selected as suggested. We have split figure 3 into two figures to try to improve the crowding. We have also done our best to move pieces of data to the supplement where appropriate (in particular for validation of tools used in the manuscript- see Supp. Fig. 3); however, we believe certain pieces of data that show the effect of TRAM2 or SEL1L on collagen or MRC2 are relevant to the main thrust of our manuscript and thus would like to include these in the main figures.

Reviewer 4

- 1- “There is no information regarding the donors of IPF and of control lungs. All information regarding demographics and clinical data of donors should be included in a Table.”

Response: Please see expanded Methods section and Supp. Table 3 (and response to reviewer 3 point 3)

- 2- *“How many IPF and control lung tissues were used per experiment? In Figure 8 it is indicated that N=7 for the western blot but it is not clear if this is also the case for the immunofluorescence and for the PCR experiments.”*

Response: We have clarified this in the figure legend and text. We were able to obtain additional tissue for protein and immunofluorescence analysis and expand the N as described; however we were not able to expand the N for the QPCR experiment and think this experiment might be underpowered relative to the protein analysis experiments; therefore we have removed the QPCR results from the manuscript and will address the mRNA levels of SEL1L in IPF vs. control lungs in future studies.

- 3- *“There is no description of the procedure used to isolate and establish primary cultures of human lung fibroblasts. From how many donors was this performed? Why the authors did not grow primary fibroblasts from IPF lung tissues as well?”*

Response: As in our response to reviewer 3 point 3, this Methods heading was a typo as we did not include primarily isolated fibroblasts in this manuscript -it has been corrected.

- 4- *“The authors should also provide the reasons why they have used the type of cells that they used in their study.”*

Response: We have clarified the reasons for use of various cell types in the text of the manuscript (see response to reviewer 3, points 3 and 6).

- 5- *“The discussion of the manuscript is very short. The authors should put their results in context to IPF and other fibrotic diseases that may share similar pathophysiological mechanisms and explain how the results presented in this manuscript can be used in a clinical setting. The limitations of the study should also be discussed in a paragraph of the discussion.”*

Response: We have expanded the Discussion section as suggested.

We believe we have addressed all reviewer and editorial concerns and suggestions. We are grateful for the opportunity to submit a revision and look forward to additional feedback.

Sincerely,
Kamran Atabai, MD

Michael Podolsky, MD

REVIEWER COMMENTS

Reviewer #1 (Remarks to the Author):

Comments for authors, Revised Manuscript August 2023:

The following comments refer to the revisions made and (in quotation marks) the statements in the author's rebuttal letter. Please refer to the previous review from this reviewer regarding the numbering of major and minor points.

General comment:

In the revised manuscript, the authors have strengthened the study. The concerns remaining are mostly connected with some partial conclusions which appear a bit too firm. This, however, does not invalidate the over-all value of the findings and may be corrected by adjustments in the text.

In brief, in the model put forward, the cellular synthesis/secretion of collagen is monitored by the ER protein, SEL1L, which leads to upregulation of the endocytic collagen receptor uPARAP/Endo180, thus increasing the cellular uptake capacity for collagen fragments in a negative feed-back mechanism. In its present form, the paper provides convincing evidence for, 1) stimulation of uPARAP/Endo180 expression and collagen uptake by collagen synthesis/secretion, and 2) uPARAP/Endo180 expression and collagen uptake being (in part) dependent on SEL1L. Furthermore, results are presented to show a molecular (FN2-dependent) interaction between SEL1L and collagen, as well as a role of the SEL1L FN2-domain in the above-described composite mechanism. The latter results are taken to substantiate the last element in the proposed mechanism, namely that it is the direct interaction of biosynthetic collagen with SEL1 that drives the stimulatory process.

In my opinion, the observed downstream effects of collagen synthesis/secretion are highly interesting in their own right, whereas the above-mentioned role of the molecular interaction cannot be taken as a proven fact based on the results available. Rather, the latter part contributes to an interesting hypothesis. Therefore, I suggest to moderate the discussion regarding the certainty of the causative role of the molecular interaction between SEL1L and collagen; see also major point 5, below.

To the specific points addressed:

Major points:

1) Quantification, western blots: "We have added densitometry of the multiple blots referenced for each experiment to clarify the findings".

This does indeed provide support to the observations, although in some cases the densitometry measurements suggest that the regulatory consequences are quite modest (Fig. 5D and Suppl. Fig. 3F). This is noted here, not to invalidate the findings, but to underscore that the proposed mechanism may not be quite as dominant as the text might suggest. This, again, is connected with major point 5, below, calling for some moderation of the conclusions. Furthermore, it is confusing that in some cases the western blot illustrations show beta-actin as the loading control while the densitometry panels refer to GAPDH (Fig. 3E/F, 5C/D and Suppl. Fig.3B/C). Is the latter designation a typo, or has a separate

densitometry measurement been performed with GAPDH ? Finally, it would have been nice to also include densitometry in the new Fig. 7G and in Fig. 7E.

2) Study based on U937 (leukemic) cells: “To address this concern, we have directly silenced COL1A1 in MRC5 fibroblasts and show that this leads to a reduction in collagen uptake and MRC2 protein, as expected based on our findings in U937 cells (Fig. 3L-N)”.

This is an important improvement of the paper. Was a similar silencing of SEL1L (not just overexpression) tested in these cells ? Although the authors present reasonable technical arguments for the use of different cell types in different parts of the study (U937, MRC5, MEFs and HEK cells), this makes it difficult to prove that the entire mechanism is active in one given cell type.

3) Inducible knockout mouse embryonic fibroblast cell line; description of these cells and the control cells, littermate controls, etc.: “We have added details on this in the Methods section.”

This documentation is important and satisfactory.

4) Collagen interaction with the FN2 domain of SEL1L, domain deletion mutant: “We have added multiple new pieces of data to address this concern. We show computational prediction of docking of collagen peptide into the FN2 domain (Supp. Fig. 4) and new additional experimental evidence of interaction based on co-IP of collagen and purified WT vs. FN2 mutant (Supp. Fig. 5). To address concerns about the folding and stability of the mutant, we show computational prediction of nearly complete overlap between the full length and mutant (Supp. Fig. 5), showing no change in remaining protein structure is expected based on the RoseTTAFold engine. In addition, we show that the mutant construct is translated and is of the expected molecular weight and that the full length and mutant proteins both co-localize with the ER marker Calnexin, showing that the mutant is in the right subcellular localization (Supp. Fig. 5). Finally, we present circular dichroism spectra of the purified full length and mutant proteins that are nearly overlapping, suggesting no changes in secondary structure caused by the mutant construct (Supp. Fig. 5).”

Although these additions are valuable, in my opinion more direct evidence would still be necessary if the authors want to make a strong point about the presumed interaction between collagen and the FN2 domain in a cellular setting. For example, computational modeling is interesting and suggestive but not experimental evidence. Co-IP studies may tell about an interaction but not about affinity and only little about specificity (unless the domain deletion mutant is considered a complete proof, which I still find disputable). The latter reservation about co-IP is particularly crucial with collagen as the interaction partner because this protein is notoriously sticky. This is not to raise doubt that some interaction between collagen and (the FN2-domain of) SEL1L can be demonstrated under certain conditions, but rather to question the certainty that this interaction is responsible for the proposed cellular mechanism. Referring to my initial comments (to the first version of the ms), I still consider that the hypothesis of the FN2 domain of SEL1L interacting with collagen (i.e., inside the cell, within the ER) has not been rigorously verified. Specifically, in my opinion the sum of the studies (biochemical as well as cell-based) is still insufficient to prove that, in a cellular setting, biosynthetic collagen would interact with SEL1L to enable the proposed mechanism (see general comment above). This is the case even though some of the cellular studies include the FN2 deletion mutant.

5) Exaggerated conclusions that can be drawn from a limited number of observations: “We have updated the language used in the manuscript as suggested.”

Although this is appreciated, it is still my opinion that some conclusions are too square when compared with the experimental evidence. In particular, this is the case for the mechanistic role of the molecular interaction between SEL1L and collagen (see above). Thus, the Results section includes the statement: ... Taken all together these data strongly suggest that SEL1L is an internal sensor of collagen biosynthesis, and SEL1L is necessary for the effect of collagen biosynthesis on MRC2 message levels and hence the downstream phenotype of collagen uptake ... (p. 9-10, line 228-230). Even if the authors consider the evidence sufficient to say "strongly suggest", then in any case it is not justified to present the same model later on as a fact that has now been proven (Discussion: SEL1L is an internal sensor of collagen biosynthesis and its collagen sensing function is necessary for the homeostatic effect of collagen biosynthesis on upregulation of MRC2 ... (p. 11, line 273-274). The same is the case for the wording used in the heading of Fig. 7.

Minor points:

1) Gene Ontology terms and phagocytosis: “We appreciate that MRC2-mediated collagen uptake is not a phagocytic process, but some genes identified as hits in our uptake screen were also identified as hits in the referenced cargo uptake screens (e.g. RAB1A and INPP5D). We therefore believe the data in this figure showing specificity of the highlighted pathways to the collagen uptake screen has some relevance (our screen was a phenotypic screen for fluorescent collagen uptake, which at least in theory could be taken up by phagocytic pathways as well as non-phagocytic pathways).”

Although speculative, I agree this cannot be excluded. A short description may be added to present this hypothesis.

2) Selection of top genes from the Gene Ontology gene set: “We have clarified how the genes from 3A were selected in the legend as suggested.”

This has still been done to a minimum only; addition of a few numbers / quantitative selection criteria would be helpful.

3) Minor points 3 and 5, typos: These have been corrected.

4) Strong fluorescence signal in Fig. 6A, both channels: “We have added additional images with lower brightness (Supp. Fig. 4B).”

This is a nice addition. Actually, to me this appears more informative than the main figure. The authors may consider to include it directly in Fig. 7.

Reviewer #2 (Remarks to the Author):

Reviewer #2

Major point 1.

The authors presented additional data on binding experiments using purified SEL1, revealing a direct interaction between the FN2 domain of SEL1 and collagen. However, the purity of the recombinant SEL1

was not determined. Depending on purity, the possibility of binding through impurities cannot be ruled out. The authors must show purity of the purified SEL1 was determined using CBB staining.

Major point 2.3.

Additional data showed that the FN2 domain of SEL1 could recognize collagen and enhance MRC2 mRNA expression. However, how SEL1 recognizes collagen and triggers MRC2 transcription is not yet known. Even though the idea that SEL1 senses collagen levels independent of ERAD is impressive, sufficient evidence to show that it is an ERAD-independent phenomenon has not been presented. To rule out any indirect effects of ERAD substrates of SEL1 on MRC2 regulation, we strongly recommend performing experiments with proteasome inhibitors. Furthermore, as shown in Fig. 7G, OS9 was used as an ERAD substrate. However, other studies have suggested that OS9 is a major component of ERAD, which may lead to confusion. It may be better to evaluate this using a typical ERAD substrate. In Fig7G, it's difficult to conclude whether this effect is due to protein degradation or protein synthesis since they only confirmed OS9 expression,

Reviewer #4 (Remarks to the Author):

In their revised manuscript, the authors have successfully addressed the points raised by this reviewer.

Response to reviews

We again thank the editor and reviewers for their thoughtful comments and suggestions. We have attempted to address all additional comments and concerns both with new experiments and in the text of the manuscript and believe these changes have again significantly strengthened the manuscript. Specifically, here is a point-by-point response to the reviews:

Reviewer 1:

Major points:

- 1- *'Quantification, western blots: "We have added densitometry of the multiple blots referenced for each experiment to clarify the findings". This does indeed provide support to the observations, although in some cases the densitometry measurements suggest that the regulatory consequences are quite modest (Fig. 5D and Suppl. Fig. 3F). This is noted here, not to invalidate the findings, but to underscore that the proposed mechanism may not be quite as dominant as the text might suggest. This, again, is connected with major point 5, below, calling for some moderation of the conclusions. Furthermore, it is confusing that in some cases the western blot illustrations show beta-actin as the loading control while the densitometry panels refer to GAPDH (Fig. 3E/F, 5C/D and Suppl. Fig. 3B/C). Is the latter designation a typo, or has a separate densitometry measurement been performed with GAPDH? Finally, it would have been nice to also include densitometry in the new Fig. 7G and in Fig. 7E.'*

Response: The discrepancy between illustrations shown and densitometry was a typo – we have now corrected this. We have added densitometry for the other panels as requested.

- 2- *'Study based on U937 (leukemic) cells: "To address this concern, we have directly silenced COL1A1 in MRC5 fibroblasts and show that this leads to a reduction in collagen uptake and MRC2 protein, as expected based on our findings in U937 cells (Fig. 3L-N)". This is an important improvement of the paper. Was a similar silencing of SEL1L (not just overexpression) tested in these cells? Although the authors present reasonable technical arguments for the use of different cell types in different parts of the study (U937, MRC5, MEFs and HEK cells), this makes it difficult to prove that the entire mechanism is active in one given cell type.'*

Response: We had not done this in the original manuscript, but have now performed these experiments and added this (See Fig. 5) – results are consistent with the data in the rest of the manuscript. We thank the reviewer for the suggestion.

- 3- *'Inducible knockout mouse embryonic fibroblast cell line; description of these cells and the control cells, littermate controls, etc.: "We have added details on this in the Methods section."*

This documentation is important and satisfactory.'

Response: We thank the reviewer for the suggestion.

- 4- *'Collagen interaction with the FN2 domain of SEL1L, domain deletion mutant: "We have added multiple new pieces of data to address this concern. We show computational prediction of docking of collagen peptide into the FN2 domain (Supp. Fig. 4) and new additional experimental evidence of interaction based on co-IP of collagen and purified WT vs. FN2 mutant (Supp. Fig. 5). To address concerns about the folding and stability of the mutant, we show computational prediction of nearly complete overlap between the full length and mutant (Supp. Fig. 5), showing no change in remaining protein structure is expected based on the RoseTTAFold engine. In addition, we show that the mutant*

construct is translated and is of the expected molecular weight and that the full length and mutant proteins both co-localize with the ER marker Calnexin, showing that the mutant is in the right subcellular localization (Supp. Fig. 5). Finally, we present circular dichroism spectra of the purified full length and mutant proteins that are nearly overlapping, suggesting no changes in secondary structure caused by the mutant construct (Supp. Fig. 5)."

Although these additions are valuable, in my opinion more direct evidence would still be necessary if the authors want to make a strong point about the presumed interaction between collagen and the FN2 domain in a cellular setting. For example, computational modeling is interesting and suggestive but not experimental evidence. Co-IP studies may tell about an interaction but not about affinity and only little about specificity (unless the domain deletion mutant is considered a complete proof, which I still find disputable). The latter reservation about co-IP is particularly crucial with collagen as the interaction partner because this protein is notoriously sticky. This is not to raise doubt that some interaction between collagen and (the FN2-domain of) SEL1L can be demonstrated under certain conditions, but rather to question the certainty that this interaction is responsible for the proposed cellular mechanism. Referring to my initial comments (to the first version of the ms), I still consider that the hypothesis of the FN2 domain of SEL1L interacting with collagen (i.e., inside the cell, within the ER) has not been rigorously verified. Specifically, in my opinion the sum of the studies (biochemical as well as cell-based) is still insufficient to prove that, in a cellular setting, biosynthetic collagen would interact with SEL1L to enable the proposed mechanism (see general comment above). This is the case even though some of the cellular studies include the FN2 deletion mutant.'

Response: We thank the reviewer and agree; we have modified language in the results and discussion sections as appropriate to emphasize the provisional nature of some of the conclusions that were stated more strongly in the prior versions of this manuscript.

- 5- *'Exaggerated conclusions that can drawn from a limited number of observations: "We have updated the language used in the manuscript as suggested."*

Although this is appreciated, it is still my opinion that some conclusions are too square when compared with the experimental evidence. In particular, this is the case for the mechanistic role of the molecular interaction between SEL1L and collagen (see above). Thus, the Results section includes the statement: ... Taken all together these data strongly suggest that SEL1L is an internal sensor of collagen biosynthesis, and SEL1L is necessary for the effect of collagen biosynthesis on MRC2 message levels and hence the downstream phenotype of collagen uptake ... (p. 9-10, line 228-230). Even if the authors consider the evidence sufficient to say "strongly suggest", then in any case it is not justified to present the same model later on as a fact that has now been proven (Discussion: SEL1L is an internal sensor of collagen biosynthesis and its collagen sensing function is necessary for the homeostatic effect of collagen biosynthesis on upregulation of MRC2 ... (p. 11, line 273-274). The same is the case for the wording used in the heading of Fig. 7.'

Response: We have updated the language used in the manuscript as suggested (see also response to point 4 above).

Minor points:

- 1- *'Gene Ontology terms and phagocytosis: "We appreciate that MRC2-mediated collagen uptake is not a phagocytic process, but some genes identified as hits in our uptake screen were also identified as hits in the referenced cargo uptake screens (e.g. RAB1A and INPP5D). We therefore believe the data in this figure showing specificity of the highlighted pathways to the collagen uptake screen has some relevance (our screen was a phenotypic screen for fluorescent collagen uptake, which at least in theory could*

be taken up by phagocytic pathways as well as non-phagocytic pathways)."
Although speculative, I agree this cannot be excluded. A short description may be added to present this hypothesis.'

Response: We have added a short description as suggested.

- 2- *'Selection of top genes from the Gene Ontology gene set: "We have clarified how the genes from 3A were selected in the legend as suggested."*

This has still been done to a minimum only; addition of a few numbers / quantitative selection criteria would be helpful.'

Response: Thank you for pointing this out. A prior version of this heatmap was accidentally used that was missing a few genes, which we have now corrected; we have updated this heatmap and added more information as requested.

- 3- *'Minor points 3 and 5, typos: These have been corrected.'*

Response: We thank the reviewer for the suggestion.

- 4- *'Strong fluorescence signal in Fig. 6A, both channels: "We have added additional images with lower brightness (Supp. Fig. 4B)."*

This is a nice addition. Actually, to me this appears more informative than the main figure. The authors may consider to include it directly in Fig. 7.'

Response: Thank you for the suggestion – we have swapped these panels in the revision.

Reviewer 2:

- 1- *'The authors presented additional data on binding experiments using purified SEL1, revealing a direct interaction between the FN2 domain of SEL1 and collagen. However, the purity of the recombinant SEL1 was not determined. Depending on purity, the possibility of binding through impurities cannot be ruled out. The authors must show purity of the purified SEL1 was determined using CBB staining.'*

Response: We have included CBB staining (Supp. Fig. 5C). Though faint, there seem to be other bands present at high molecular weight and potentially elsewhere (a faint band at ~60kDa may be endogenous MYC). Therefore we agree that the purity of the isolated proteins cannot be claimed to be complete and a direct molecular interaction cannot be concluded with certainty (as pointed out by reviewer 1 – see major point 4 and response). We have made sure to include language in the Results and Discussion to describe the limitations of our approach.

- 2- *'Additional data showed that the FN2 domain of SEL1 could recognize collagen and enhance MRC2 mRNA expression. However, how SEL1 recognizes collagen and triggers MRC2 transcription is not yet known. Even though the idea that SEL1 senses collagen levels independent of ERAD is impressive, sufficient evidence to show that it is an ERAD-independent phenomenon has not been presented. To rule out any indirect effects of ERAD substrates of SEL1 on MRC2 regulation, we strongly recommend performing experiments with proteasome inhibitors. Furthermore, as shown in Fig. 7G, OS9 was used as an ERAD substrate. However, other studies have suggested that OS9 is a major component of ERAD, which may lead to confusion. It may be better to evaluate this using a typical ERAD substrate. In Fig7G, it's difficult to conclude whether this effect is due to protein degradation or protein synthesis since they only confirmed OS9 expression'*

Response: We thank the reviewer for these suggestions. To address these points, we have performed additional experiments to look at stability of MRC2 protein and to look at ERAD function in the context of KO of SEL1L or in-frame deletion of the FN2 domain. We have done this using new cell lines with CRISPR-mediated in-frame deletion of FN2 domain (described in the methods) to provide additional orthogonal validation of our overexpression constructs (these new cell lines have endogenous expression levels of

SEL1L, avoiding any confounding by high levels of overexpressed proteins, or any consequences of viral vectors). See Fig. 8 for details. We have performed experiments with cycloheximide and proteasome inhibitor MG-132 as suggested by the reviewer. In these data we show that KO of SEL1L leads to accumulation of OS9 as expected and this is due to decreased protein degradation (consistent with this being an ERAD substrate, as previously shown). However, OS9 levels are restored in the FN2 deletion mutant. These are consistent with and add to the data we showed in the last version of our manuscript. Additionally, we show that MRC2 protein, though lower in both the SEL1L KO and FN2 deletion mutants, does undergo some degradation in the FN2 deletion mutant after CHX but not the SEL1L KO condition – these data are consistent with the idea that the function of the FN2 domain is independent of ERAD since there is no effect on OS9 degradation or MRC2 degradation in that condition. These data also suggest that MRC2 protein may be an ERAD substrate (though these data do not prove this); but that the downregulation of MRC2 by either SEL1L KO or FN2 deletion are therefore likely not ERAD-dependent. In our opinion, the positive regulation by SEL1L is likely via mRNA transcriptional regulation, whereas the effect of ERAD on MRC2 protein levels would be expected to lower MRC2 protein levels by degradation (conversely SEL1L KO, based purely on its ERAD function, would be expected to lead to decreased degradation of MRC2 protein and hence higher levels – not what we have observed). This may in fact explain some of our data showing that HRD1 knockout seems to increase to some extent MRC2 protein levels; this is something we intend to pursue in the future. However, although we do not prove this concept, we believe our new data show that the FN2 domain function on *MRC2* expression is not via ERAD; we have emphasized these new findings in the manuscript results and discussion sections, as well as limitations. We thank the reviewer for these suggestions.

We believe we have addressed all reviewer and editorial concerns and suggestions. We are grateful for the opportunity to submit a revision and look forward to additional feedback.

Sincerely,
Kamran Atabai, MD

Michael Podolsky, MD

REVIEWER COMMENTS

Reviewer #1 (Remarks to the Author):

All of the points raised in my previous review comments have now been addressed and I consider the study is now complete.

Reviewer #2 (Remarks to the Author):

In our previous review comments, we raised two major points. I believe that these suggestions are important for ensuring the credibility of their conclusions. They answered our comments and performed some experiments, but they did not provide substantive answers. In particular, CBB staining may have misidentified the SEL1L band.

Regarding comment 1, they showed the purity of purified SEL1L using CBB staining, but there are no data to confirm that it is SEL1L. A band shift was observed in WT and FN2-deficient cells by Western Blotting, but no such band pattern was observed in CBB cells. There were two unidentified bands above the band size at which SEL1L was detected, indicating that most of the purified products were unidentified proteins. The attached figure shows the two unidentified bands and the positions at which SEL1L should be detected.

Furthermore, in comment 2, we suggested that substrates other than OS-9 be used to check their effects on degradation. However, they again proceeded to the experiment with only OS-9 and did not respond to our comments. Another problem is that the mechanism by which SEL1L regulates MRC2 transcription is unclear, and there is no explanation for how transcription is regulated. The most likely explanation is that SEL1L's degradation substrates may affect MRC2 mRNA. Unless this possibility is excluded, the claim that SEL1L regulates MRC2 mRNA expression in an ERAD-independent manner cannot be accepted.

These weaken the original claims, such as direct collagen recognition via the FN2 domain of SEL1L and a mechanism for regulating MRC2 expression by SEL1L, independent of ERAD. I took seriously that these fixes of the manuscript (including the fixes for Reviewer#1) made the paper less impactful than initially intended and that it was, therefore, unsuitable for publication in Nature Communications.

C

WCL

Purified

WT

 Δ FN2

WT

 Δ FN2

(kDa)

ladder

170

130

100

70

55

40

35

25

15

← ?
← ?
| SEL1L

← Dye front

Response to reviews

We thank the editor and reviewer for their thoughtful comments and suggestions. We have made every attempt to address the additional comments and concerns both with new experiments and in the text of the manuscript. We believe these changes have significantly strengthened the manuscript. Specifically, here is a point-by-point response to the reviews:

Reviewer 2:

Major points:

- 1- *'Regarding comment 1, they showed the purity of purified SEL1L using CBB staining, but there are no data to confirm that it is SEL1L. A band shift was observed in WT and FN2-deficient cells by Western Blotting, but no such band pattern was observed in CBB cells. There were two unidentified bands above the band size at which SEL1L was detected, indicating that most of the purified products were unidentified proteins. The attached figure shows the two unidentified bands and the positions at which SEL1L should be detected.'*

Response: We thank the reviewer for the astute comments and observations. The two darkest bands in the original Coomassie stained gel image we included in the last revision and which were highlighted by the reviewer were not SEL1L (which we confirmed by proteomics of the excised gel bands; data not included in the manuscript); SEL1L is at a lower molecular weight and is more readily seen on a different CBB image which we have replaced the prior image with (see Supp. Fig. 5C – the bands marked with arrowheads were sequenced and verified to be SEL1L – see new Supp. Table 3). The reviewer is correct that the SEL1L bands were present at much lower abundance than the neighboring bands seen after protein purification via anti-MYC beads and that we therefore cannot absolutely conclude direct binding between SEL1L and collagen.

In order to further address this concern, we have re-purified protein using a second tag (FLAG) expressed on this protein, i.e. by using an anti-FLAG antibody. This isolation had relatively fewer impurities with the band at the molecular weight of SEL1L now being the most prominent band (Supp. Fig. 5G). Our findings with protein from this FLAG-tagged based isolation, as with the previous MYC-tagged based isolation (Supp. Fig. 5C), show collagen binding by the recombinant protein that is markedly greater in the WT SEL1L as compared with the construct lacking the FN2 domain (see Supp. Fig. 5G-H).

We respectfully believe that in total our data indicate that SEL1L associates with collagen either through a direct interaction or as part of a complex and that the FN2 domain of SEL1L is necessary for this interaction which subsequently leads to modulation of *MRC2* expression. In this revision, we have modified the text to reflect that this interaction may be direct or as part of a protein complex. We further believe that this series of experiments address the reviewer's prior concerns about the possibility that changes in collagen synthesis could have led to the differences in co-IP results when evaluating SEL1L WT or mutant interaction with collagen in the cell-based assay; these cell-free experiments utilize exogenously added rat-tail collagen and therefore argue against this possibility.

- 2- *'Furthermore, in comment 2, we suggested that substrates other than OS-9 be used to check their effects on degradation. However, they again proceeded to the experiment with only OS-9 and did not respond to our comments. Another problem is that the mechanism by which SEL1L*

regulates MRC2 transcription is unclear, and there is no explanation for how transcription is regulated. The most likely explanation is that SEL1L's degradation substrates may affect MRC2 mRNA. Unless this possibility is excluded, the claim that SEL1L regulates MRC2 mRNA expression in an ERAD-independent manner cannot be accepted.'

Response: We apologize for not having evaluated additional substrates on our previous resubmission. We have now, as suggested, tested other known endogenous ERAD substrates (CD147, SHH, and IRE1 α) and all follow the same pattern as OS9 (see Fig. 8F). We also reference our co-authors' recently accepted publication (Weis Q et al...Qi L. *JCI* 2023) which shows that the FN2 domain of SEL1L is dispensable for the ERAD function of SEL1L. Our new data with additional ERAD substrates coupled with the recently published work (Weis Q et al...Qi L. *JCI* 2023) demonstrating that the FN2 domain does not regulate ERAD function leads us to conclude that ERAD is preserved in the FN2-deletion mutant. We therefore believe that the mechanism by which *MRC2* is regulated through SEL1L in an FN2 domain-dependent manner is predominantly if not completely independent of ERAD. These data also support the functionality of the FN2-deletion mutant thereby arguing against the possibility of protein misfolding contributing to differential binding of SEL1L or its interactors to collagen.

We do hope that we have adequately addressed the reviewers' concerns.

Sincerely,
Kamran Atabai, MD

Michael Podolsky, MD

REVIEWERS' COMMENTS

Reviewer #2 (Remarks to the Author):

I have reviewed the updated data, and it indeed satisfies my requirements. Thank you for accommodating my request.